# A Deeper Look at Discounting Mismatch in Actor-Critic Algorithms

## Abstract

We investigate the discounting mismatch in actor-critic algorithm implementations from a representation learning perspective. Theoretically, actor-critic algorithms usually have discounting for both actor and critic, *i.e.*, there is a $\gamma^t$ term in the actor update for the transition observed at time $t$ in a trajectory and the critic is a discounted value function. Practitioners, however, usually ignore the discounting ($\gamma^t$) for the actor while using a discounted critic. We investigate this mismatch in two scenarios. In the first scenario, we consider optimizing an undiscounted objective ($\gamma = 1$) where $\gamma^t$ disappears naturally ($1^t = 1$). We then propose to interpret the discounting in critic in terms of a *bias-variance-representation* trade-off and provide supporting empirical results. In the second scenario, we consider optimizing a discounted objective ($\gamma < 1$) and propose to interpret the omission of the discounting in the actor update from an *auxiliary task* perspective and provide supporting empirical results.

## 1 Introduction

Actor-critic algorithms have enjoyed great success both theoretically (Williams, 1992; Sutton et al., 2000; Konda, 2002; Schulman et al., 2015a) and empirically (Mnih et al., 2016; Silver et al., 2016; Schulman et al., 2017; OpenAI, 2018). There is, however, a longstanding gap between the theory behind actor-critic algorithms and how practitioners implement them. Let $\gamma, \gamma_A$, and $\gamma_C$ be the discount factors for defining the objective, updating the actor, and updating the critic respectively. Theoretically, no matter whether $\gamma = 1$ or $\gamma < 1$, we should always use $\gamma_A = \gamma_C = \gamma$ (Sutton et al., 2000; Schulman et al., 2015a) or at least keep $\gamma_A = \gamma_C$ if Blackwell optimality (Veinott, 1969; Weitzman, 2001) [1] is considered. Practitioners, however, usually use $\gamma_A = 1$ and $\gamma_C < 1$ in their implementations (Dhariwal et al., 2017; Caspi et al., 2017; Zhang, 2018; Kostrikov, 2018; Achiam, 2018; Liang et al., 2018; Stooke & Abbeel, 2019). Although this mismatch and its theoretical disadvantage have been recognized by Thomas (2014); Nota & Thomas (2020), whether and why it yields benefits in practice has not been systematically studied. In this paper, we empirically investigate this mismatch from a representation learning perspective. We consider two scenarios separately.

**Scenario 1:** *The true objective is undiscounted ($\gamma = 1$).* The theory prescribes to use $\gamma_A = \gamma_C = \gamma = 1$. Practitioners, however, usually use $\gamma_A = \gamma = 1$ but $\gamma_C < 1$, introducing *bias*. We explain this mismatch with the following hypothesis:

**Hypothesis 1.** $\gamma_C < 1$ *optimizes a bias-variance-representation trade-off.*

It is easy to see that $\gamma_C < 1$ reduces the variance in bootstrapping targets. Besides this, we further provide empirical evidence showing that when $\gamma_C < 1$, it may become easier to find a good representation compared to $\gamma_C = 1$. Consequently, although using $\gamma_C < 1$ introduces bias, it can facilitate representation learning. For our empirical study, we make use of recently introduced techniques, such fixed horizon temporal different learning (De Asis et al., 2019) and distributional reinforcement learning (Bellemare et al., 2017) to disentangle the various effects the discount factor has on the learning process.

**Scenario 2:** *The true objective function is discounted ($\gamma < 1$).* Theoretically, there is a $\gamma^t$ term for the actor update on a transition observed at time $t$ in a trajectory (Sutton et al., 2000; Schulman

---

[1] Blackwell optimality states that, in finite MDPs, there exists a $\gamma_0 < 1$ such that for all $\gamma \geq \gamma_0$, the optimal policies for the $\gamma$-discounted objective are the same.

et al., 2015a). Practitioners, however, usually ignore this term while using a discounted critic, *i.e.*, $\gamma_A = 1$ and $\gamma_C = \gamma < 1$ are used. We explain this mismatch with the following hypothesis:

**Hypothesis 2.** *Using $\gamma_C = \gamma < 1$ and $\gamma_A = 1$ is effectively similar to using $\gamma_C = \gamma_A = \gamma < 1$ plus an auxiliary loss that sometimes facilitates representation learning.*

Our empirical study involves implementing the auxiliary task explicitly by using an additional policy for optimizing the difference term between the loss of $\gamma_A = 1$ and the loss of $\gamma_A < 1$. We also design new benchmarking environments where the sign of the reward function is flipped after a certain time step such that later transitions differ from earlier ones. In that setting, $\gamma_A = 1$ becomes harmful.

## 2 BACKGROUND

**Markov Decision Processes:** We consider an infinite horizon MDP with a finite state space $\mathcal{S}$, a finite action space $\mathcal{A}$, a bounded reward function $r : \mathcal{S} \to \mathbb{R}$, a transition kernel $p : \mathcal{S} \times \mathcal{S} \times \mathcal{A} \to [0,1]$, an initial state distribution $\mu_0$, and a discount factor $\gamma \in [0,1]$.[2] The initial state $S_0$ is sampled from $\mu_0$. At time step $t$, an agent in state $S_t$ takes action $A_t \sim \pi(\cdot|S_t)$, where $\pi : \mathcal{A} \times \mathcal{S} \to [0,1]$ is the policy it follows. The agent then gets a reward $R_{t+1} \doteq r(S_t)$ and proceeds

| $\gamma$ | define the objective |
|---|---|
| $\gamma_A$ | update the actor |
| $\gamma_C$ | update the critic |

Table 1: Roles of the different discount factors

to the next state $S_{t+1} \sim p(\cdot|S_t, A_t)$. The return of the policy $\pi$ at time step $t$ is defined as $G_t \doteq \sum_{i=1}^{\infty} \gamma^{i-1} R_{t+i}$, which allows us to define the state value function $v_\pi^\gamma(S) \doteq \mathbb{E}[G_t|S_t = s]$ and the state-action value function $q_\pi^\gamma(s,a) \doteq \mathbb{E}[G_t|S_t = s, A_t = a]$. We consider episodic tasks where we assume there is an absorbing state $s^\infty \in \mathcal{S}$ such that $r(s^\infty) = 0$ and $p(s^\infty|s^\infty, a) = 1$ holds for any $a \in \mathcal{A}$. When $\gamma < 1$, $v_\pi^\gamma$ and $q_\pi^\gamma$ are always well defined. When $\gamma = 1$, to ensure $v_\pi^\gamma$ and $q_\pi^\gamma$ are well defined, we further assume finite expected episode length. Let $T_s^\pi$ be a random variable denoting the first time step that an agent hits $s^\infty$ when following $\pi$ given $S_0 = s$. We assume $T_{\max} \doteq \sup_{\pi \in \Pi} \max_s \mathbb{E}[T_s^\pi] < \infty$, where $\pi$ is parameterized by $\theta$ and $\Pi$ is the corresponding function class. Similar assumptions are also used in stochastic shortest path problems (*e.g.*, Section 2.2 of Bertsekas & Tsitsiklis (1996)). In our experiments, all the environments have a hard time limit of 1000, i.e., $T_{\max} = 1000$. This is standard practice, classic RL environments also have an upper limit on their episode lengths (*e.g.* 27k in Bellemare et al. (2013, ALE)). Following Pardo et al. (2018), we add the (normalized) time step $t$ in the state to keep the environment Markovian. We measure the performance of a policy $\pi$ with $J_\gamma(\pi) \doteq \mathbb{E}_{S_0 \sim \mu_0}[v_\pi^\gamma(S_0)]$.

**Vanilla Policy Gradient:** Sutton et al. (2000) compute $\nabla_\theta J_\gamma(\pi)$ as

$$\nabla_\theta J_\gamma(\pi) \doteq \sum_s d_\pi^\gamma(s) \sum_a q_\pi^\gamma(s,a) \nabla_\theta \pi(a|s), \tag{1}$$

where $d_\pi^\gamma(s) \doteq \sum_{t=0}^{\infty} \gamma^t \Pr(S_t = s|\mu_0, p, \pi)$ for $\gamma < 1$ and $d_\pi^\gamma(s) \doteq \mathbb{E}[\sum_{t=0}^{T_{S_0}^\pi} \Pr(S_t = s|S_0, p, \pi)]$ for $\gamma = 1$.[3] Note $d_\pi^\gamma$ remains well-defined for $\gamma = 1$ when $T_{\max} < \infty$. In order to optimize the policy performance $J_\gamma(\pi)$, one can follow (1) and, at time step $t$, update $\theta_t$ as

$$\theta_{t+1} \leftarrow \theta_t + \alpha \gamma_A^t q_\pi^{\gamma_C}(S_t, A_t) \nabla_\theta \log \pi(A_t|S_t), \tag{2}$$

where $\alpha$ is a learning rate. If we replace $q_\pi^{\gamma_C}$ with a learned value function, the update rule (2) becomes an actor-critic algorithm, where the actor refers to $\pi$ and the critic refers to the learned approximation of $q_\pi^{\gamma_C}$. In practice, an estimate for $v_\pi^{\gamma_C}$ instead of $q_\pi^{\gamma_C}$ is usually learned. Theoretically, we should have $\gamma_A = \gamma_C = \gamma$. Practitioners, however, usually ignore the $\gamma_A^t$ term in (2), and use $\gamma_C < \gamma_A = 1$. What this update truly optimizes remains an open problem (Nota & Thomas, 2020).

**TRPO and PPO:** To improve the stability of actor-critic algorithms, Schulman et al. (2015a) propose Trust Region Policy Optimization (TRPO), based on the performance improvement lemma:

**Lemma 1.** *(Theorem 1 in Schulman et al. (2015a)) For $\gamma < 1$ and any two policies $\pi$ and $\pi'$,*

$$J_\gamma(\pi') \geq J_\gamma(\pi) + \left( \sum_s d_\pi^\gamma(s) \sum_a \pi'(a|s) Adv_\pi^\gamma(s,a) \right) - \frac{4 \max_{s,a} |Adv_\pi^\gamma(s,a)| \gamma \epsilon(\pi, \pi')}{(1-\gamma)^2},$$

---

[2]Following Schulman et al. (2015a), we consider $r : \mathcal{S} \to \mathbb{R}$ instead of $r : \mathcal{S} \times \mathcal{A} \to \mathbb{R}$ for simplicity.

[3]Sutton et al. (2000) do not explicitly define $d_\pi^\gamma$ when $\gamma = 1$, which, however, can be easily deduced from Chapter 13.2 in Sutton & Barto (2018).

*where* $Adv_\pi^\gamma(s, a) \doteq \mathbb{E}_{s' \sim p(\cdot|s,a)}[r(s) + \gamma v_\pi^\gamma(s') - v_\pi^\gamma(s)]$ *is the advantage,* $\epsilon(\pi, \pi') \doteq \max_s D_{KL}(\pi(\cdot|s)||\pi'(\cdot|s))$, *and* $D_{KL}$ *refers to the KL divergence.*

To facilitate our empirical study, we first make a theoretical contribution by extending Lemma 1 to the undiscounted setting. We have the following lemma:

**Lemma 2.** *Assuming* $T_{\max} < \infty$, *for* $\gamma = 1$ *and any two policies* $\pi$ *and* $\pi'$,

$$J_\gamma(\pi') \geq J_\gamma(\pi) + \Big( \sum_s d_\pi^\gamma(s) \sum_a \pi'(a|s) Adv_\pi^\gamma(s, a) \Big) - 4 \max_{s,a} |Adv_\pi^\gamma(s, a)| T_{\max}^2 \epsilon(\pi, \pi').$$

The proof of Lemma 2 is provided in the appendix. A practical implementation of Lemmas 1 and 2 is to compute a new policy $\theta$ via gradient ascent on the clipped objective:

$$L(\theta) \doteq \sum_{t=0}^\infty \gamma_A^t \min \Big\{ \frac{\pi_\theta(A_t|S_t)}{\pi_{\theta_{\text{old}}}(A_t|S_t)} \text{Adv}_{\pi_{\theta_{\text{old}}}}^{\gamma_C}(S_t, A_t), \text{clip}(\frac{\pi_\theta(A_t|S_t)}{\pi_{\theta_{\text{old}}}(A_t|S_t)}) \text{Adv}_{\pi_{\theta_{\text{old}}}}^{\gamma_C}(S_t, A_t) \Big\}, \quad (3)$$

where $S_t$ and $A_t$ are sampled from $\pi_{\theta_{\text{old}}}$, and $\text{clip}(x) \doteq \max(\min(x, 1 + \epsilon), 1 - \epsilon)$ with $\epsilon$ a hyperparameter. Theoretically, we should have $\gamma_A = \gamma_C$, but practical algorithms like Proximal Policy Optimization (Schulman et al., 2017, PPO) usually use $\gamma_C < \gamma_A = 1$.

**Policy Evaluation:** We now introduce several policy evaluation techniques we use in our empirical study. Let $\hat{v}$ be our estimate of $v_\pi^\gamma$. At time step $t$, Temporal Difference learning (TD, Sutton (1988)) updates $\hat{v}$ as $\hat{v}(S_t) \leftarrow \hat{v}(S_t) + \alpha(R_{t+1} + \gamma\hat{v}(S_{t+1}) - \hat{v}(S_t))$. Instead of the infinite horizon discounted return $G_t$, De Asis et al. (2019) propose to consider the $H$-step return $G_t^H \doteq \sum_{i=1}^H R_{t+i}$. Correspondingly, the $H$-step value function is defined as $v_\pi^H(s) \doteq \mathbb{E}[G_t^H|S_t = s]$. We let $\hat{v}^H$ be our estimate of $v_\pi^H$. At time step $t$, De Asis et al. (2019) use the following update rule to learn $\hat{v}^H$:

$$\hat{v}^i(S_t) \leftarrow \hat{v}^i(S_t) + \alpha(R_{t+1} + \hat{v}^{i-1}(S_{t+1}) - \hat{v}^i(S_t)) \quad (i = 1, \ldots H), \quad (4)$$

where $\hat{v}^0(s) \doteq 0$. In other words, to learn $\hat{v}^H$, we need to learn $\{\hat{v}^i\}_{i=1,\ldots,H}$ simultaneously. De Asis et al. (2019) call (4) *Fixed Horizon Temporal Difference* learning (FHTD).

As $G_t$ is a random variable, Bellemare et al. (2017) propose to learn its full distribution instead of its expectation only, yielding the Distributional Reinforcement Learning (RL) paradigm. They use a categorical distribution with 51 atoms uniformly distributed in $[-V_{\max}, V_{\max}]$ to approximate the distribution of $G_t$, where $V_{\max}$ is a hyperparameter. In this paper, we refer to the corresponding policy evaluation algorithm as *C51*.

**Methodology:** We consider MuJoCo robot simulation tasks from OpenAI gym (Brockman et al., 2016) as our benchmark. Given its popularity in understanding deep RL algorithms (Henderson et al., 2017; Ilyas et al., 2018; Engstrom et al., 2019; Andrychowicz et al., 2020) and designing new deep RL algorithms (Fujimoto et al., 2018; Haarnoja et al., 2018), we believe our empirical results are relevant to most practitioners.

We choose PPO, a simple yet effective and widely used algorithm, as the representative actor-critic algorithm for our empirical study. PPO is usually equipped with generalized advantage estimation (Schulman et al., 2015b, GAE), which has a tunable hyperparameter $\hat{\gamma}$. The roles of $\gamma$ and $\hat{\gamma}$ are similar. To reduce its confounding effect, we do not use GAE in our experiments, *i.e.*, the advantage estimation for our actor is simply the TD error $R_{t+1} + \gamma_C \hat{v}(S_{t+1}) - \hat{v}(S_t)$. The PPO pseudocode we follow is provided in Alg. 1 in the appendix and we refer to it as the default PPO implementation.

We use the standard architecture and optimizer across all tasks, in particular, the actor and the critic do not share layers. We conduct a thorough learning rate search in `Ant` for each algorithmic configuration (*i.e.*, a curve in a figure) and then use the same learning rate for all other tasks. When using FHTD and C51, we also include $H$ and $V_{\max}$ in the grid search. All details are provided in the appendix. We report the average episode return of the ten most recent episodes against the number of interactions with the environment. Curves are averages over ten independent runs with shaded regions indicating standard errors.

## 3  OPTIMIZING THE UNDISCOUNTED OBJECTIVE (SCENARIO 1)

When our goal is to optimize the *undiscounted* objective $J_{\gamma=1}(\pi)$, one theoretically grounded option is to use $\gamma_A = \gamma_C = \gamma = 1$. By using $\gamma_A = 1$ and $\gamma_C < 1$, practitioners introduce *bias*. We first

empirically confirm that introducing bias in this way indeed has empirical advantages. A simple first hypothesis is that $\gamma_C < 1$ leads to lower variance in Monte Carlo return bootstrapping targets than $\gamma_C = 1$, it thus optimizes a bias-variance trade-off. However, we further show that there are empirical advantages from $\gamma_C < 1$ that cannot uniquely be explained by this bias-variance trade-off, indicating that there are additional factors beyond variance. We then show empirical evidence identifying representation learning as an additional factor, leading to the *bias-variance-representation* trade-off from Hypothesis 1. All the experiments in this section use $\gamma_A = 1$.

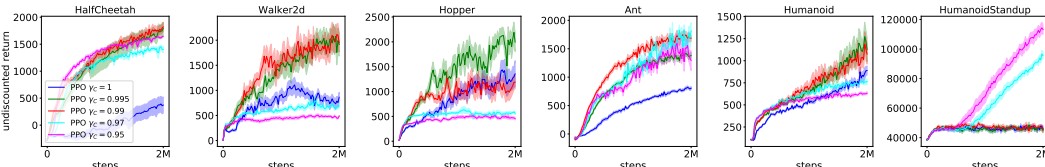

Figure 1: The default PPO implementation with different discount factors.

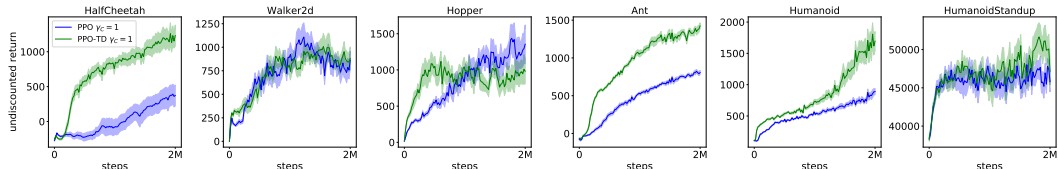

Figure 2: Comparison between PPO and PPO-TD when $\gamma_C = 1$.

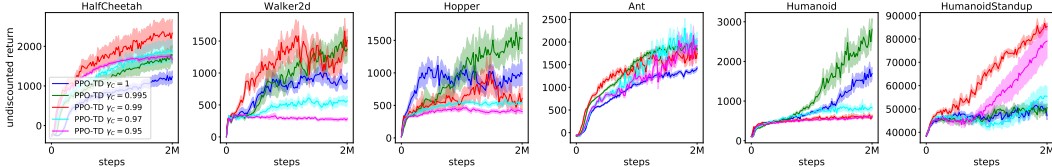

Figure 3: PPO-TD with different discount factors.

**Bias-variance trade-off:** To investigate the advantages of using $\gamma_C < 1$, we first test default PPO with $\gamma_C \in \{0.95, 0.97, 0.99, 0.995, 1\}$. We find that the best discount factor is always with $\gamma_C < 1$ and that $\gamma_C = 1$ usually leads to a performance drop (Figure 1). In default PPO, although the advantage is computed as the one-step TD error, the update target for updating the critic $\hat{v}(S_t)$ is almost always a Monte Carlo return. As there is no $\gamma_A^t$ term in the actor update, we should theoretically use $\gamma_C = \gamma_A = 1$ when computing the Monte Carlo return, which usually leads to high variance. Consequently, a simple hypothesis for the empirical advantages of using $\gamma_C < 1$ is a bias-variance trade-off. We find, however, that there is more at play.

**Beyond bias-variance trade-off:** To reduce the effect of $\gamma_C$ in controlling the variance, we benchmark PPO-TD (Algorithm 2 in the appendix). PPO-TD is the same as default PPO except that the critic is updated with one-step TD, *i.e.*, the update target for $\hat{v}(S_t)$ is now $R_{t+1} + \gamma_C \hat{v}(S_{t+1})$. Although Figure 2 shows that PPO-TD ($\gamma_C = 1$) outperforms PPO ($\gamma_C = 1$) by a large margin, indicating bias-variance may be at play, Figure 3 suggests that for PPO-TD as well, $\gamma_C < 1$ is still preferable to $\gamma_C = 1$. To further study this phenomenon, we benchmark PPO-TD-Ex (Algorithm 3 in the appendix), in which we provide $N$ extra transitions to the critic by sampling multiple actions at any single state and using an averaged bootstrapping target. The update target for $\hat{v}(S_t)$ in PPO-TD-Ex is $\frac{1}{N+1} \sum_{i=0}^{N} R_{t+1}^i + \gamma_C \hat{v}(S_{t+1}^i)$. Here $R_{t+1}^0$ and $S_{t+1}^0$ refer to the original reward and successor state. To get $R_{t+1}^i$ and $S_{t+1}^i$ for $i \in \{1, \ldots, N\}$, we first sample an action $A_t^i$ from the sampling policy, then reset the environment to $S_t$, and finally execute $A_t^i$ to get $R_{t+1}^i$ and $S_{t+1}^i$. Importantly, we do not count those $N$ extra transitions in the $x$-axis when plotting. The advantage for the actor update in PPO-TD-Ex is estimated with $R_{t+1}^0 + \hat{v}(S_{t+1}^0) - \hat{v}(S_t)$ regardless of $\gamma_C$ to further control the influence of variance. The critic $\hat{v}$ is not trained on the extra successor states $\{S_{t+1}^i\}_{i=1,\ldots,N}$.

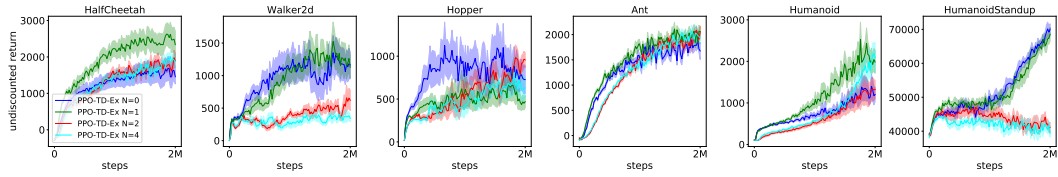

Figure 4: PPO-TD-Ex ($\gamma_c = 0.99$).

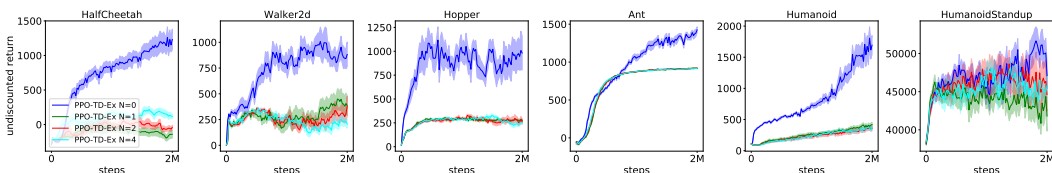

Figure 5: PPO-TD-Ex ($\gamma_c = 1$).

So the quality of the prediction $\hat{v}(S_{t+1}^i)$ depends mainly on the generalization of $\hat{v}$. Intuitively, if $\hat{v}$ generalizes well, providing proper amount of transitions this way should improve or maintain the overall performance as they help reduce variance. As shown by Figure 4, PPO-TD-Ex ($\gamma_c = 0.99$) roughly follows this intuition. However, surprisingly, providing extra data to PPO-TD-Ex ($\gamma_c = 1$) leads to a significant performance drop (Figure 5). This drop suggests that the larger variance from the randomness of $S_{t+1}$ is not the only issue when using $\gamma_c = 1$ to train the critic. The quality of the estimate $\hat{v}$, at least in terms of making prediction on untrained states $\{S_{t+1}^i\}_{1,...,N}$, is lower when $\gamma_c = 1$ is used than $\gamma_c < 1$. In other words, the generalization of $\hat{v}$ is poor when $\gamma_c = 1$. The curves for PPO-TD-Ex ($\gamma_c = 0.995$) are a mixture of $\gamma_c = 0.99$ and $\gamma_c = 1$ and are provided in Figure 16 in the appendix.

In the undiscounted setting, we should theoretically have $R_{t+1} + \hat{v}(S_{t+1})$ as the update target for the critic. When $\gamma_c < 1$ is used instead, the update target becomes $R_{t+1} + \gamma_c\hat{v}(S_{t+1})$ and the variance resulting from the randomness of $S_{t+1}$ becomes less pronounced. So here, $\gamma_c$ trades off bias with variance, similar to that in Monte Carlo return bootstrapping targets in default PPO. We refer to this effect of $\gamma_c$ as *variance control*. However, $\gamma_c$ can also affect the difficulty of learning a good estimate $\hat{v}$ for $v_\pi^{\gamma_c}$; we refer to this effect of $\gamma_c$ as *learnability control* (Lehnert et al., 2018; Laroche & van Seijen, 2018; Romoff et al., 2019). Inspired by the poor generalization of $\hat{v}$ when $\gamma_c = 1$, we investigate learnability control mainly from the representation learning perspective. By representation learning, we refer to learning the bottom layers (backbone) of a neural network. The last layer of the neural network is then interpreted as a linear function approximator whose features are the output of the backbone. This interpretation of representation learning is widely used in the RL community, see *e.g.* Jaderberg et al. (2016); Chung et al. (2018); Veeriah et al. (2019).

**Bias-representation trade-off:** To separate variance control and learnability control, ideally we should investigate the update target $R_{t+1} + \gamma_{c,1}\hat{v}(S_{t+1})$, where $\hat{v}$ is trained to approximate $v_\pi^{\gamma_c,2}$ and $\gamma_{c,2} < \gamma_{c,1} = 1$. Learning an estimate $\hat{v}$ for $v_\pi^{\gamma_c,2}$, however, implies to use the update target $R_{t+1} + \gamma_{c,2}\hat{v}(S_{t+1})$: the two effects of $\gamma_{c,2}$ then get mixed again. To solve this dilemma, we consider the update target $R_{t+1} + \hat{v}^{H-1}(S_{t+1})$, where $\hat{v}^{H-1}(S_{t+1})$ is trained to approximate $v_\pi^{H-1}$, *i.e.*, we use FHTD to train the critic in PPO, which we refer to as PPO-FHTD (Algorithm 4 in the appendix). PPO-FHTD implements $\gamma_{c,1} = 1$ directly, and manipulating $H$ changes the horizon of the policy evaluation problem, which is also one of the effects of manipulating $\gamma_{c,2}$.

We test two parameterizations for PPO-FHTD to investigate representation learning. In the first parameterization, to learn $v_\pi^H$, we parameterize $\{v_\pi^i\}_{i=1,...,H}$ as $H$ different heads over the same representation layer (backbone). In the second parameterization, we always learn $\{v_\pi^i\}_{i=1,...,1024}$ as 1024 different heads over the same representation layer, whatever $H$ we are interested in. To approximate $v_\pi^H$, we then simply use the output of the $H$-th head. A diagram (Figure 13) in the appendix further illustrates the difference between the two parameterizations.

Figure 6 shows that by tuning $H$ for FHTD, PPO-FHTD with the first parameterization matches or exceeds the performance of PPO-TD ($\gamma_c < 1$) in most tasks, and that the best $H$ is always

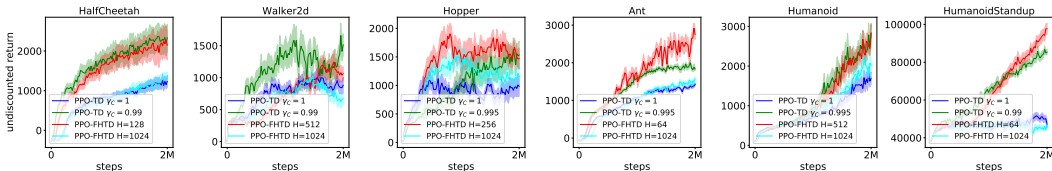

Figure 6: PPO-FHTD with the first parameterization. The best $H$ and $\gamma_C$ are used for each game.

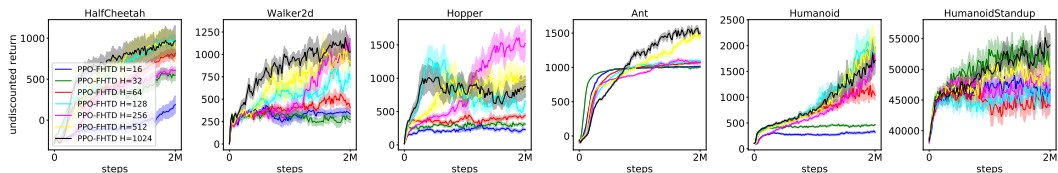

Figure 7: PPO-FHTD with the second parameterization.

smaller than 1024. Theoretically, as long as we use an $H \geq T_{\max} = 1000$, we always have $v_\pi^H(s) \equiv v_\pi^{\gamma=1}(s)$. Figure 6 shows that the performance of PPO-FHTD ($H = 1024$) is very close to PPO-TD ($\gamma_C = 1$), indicating that learning $\{v_\pi^i\}_{i=1,\dots,1023}$ is not an additional overhead for the network in terms of learning $v_\pi^{H=1024}$, *i.e.*, increasing $H$ does not pose additional challenges in terms of network capacity. However, Figure 7 suggests that for the second parameterization, $H = 1024$ is almost always among the best choices of $H$. Comparing Figures 6 and 7, we conclude that in the tested domains, learning $v_\pi^H$ with different $H$ requires different representations. This suggests that we can interpret the results in Figure 6 as a *bias-representation trade-off*. Using a larger $H$ is less biased but representation learning *may* become harder due to the longer policy evaluation horizon. Consequently, an intermediate $H$ achieves the best performance in Figure 6. As reducing $H$ cannot bring in advantages in representation learning under the second parameterization, the less biased $H$, *i.e.*, the larger $H$, usually performs better in Figure 7. Overall, $\gamma_C$ optimizes a *bias-representation* trade-off by changing the policy evaluation horizon $H$.

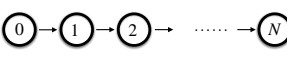

Figure 8: A simple MRP.

We further conjecture that representation learning may be harder for a longer horizon because good representations can become rarer. We provide a simulated example to support this. Consider policy evaluation on the simple Markov Reward Process (MRP) from Figure 8. We assume the reward for each transition is fixed and is randomly generated in $[0, 1]$. Let $x_s \in \mathbb{R}^K$ be the feature vector for a state $s$; we set its $i$-th component as $x_s[i] \doteq \tanh(\xi)$, where $\xi$ is a random variable uniformly distributed in $[-2, -2]$. We chose this feature setup as we use $\tanh$ as the activation function in our PPO. We use $X \in \mathbb{R}^{N \times K}$ to denote the feature matrix. To create state aliasing (McCallum, 1997), which is common under function approximation, we first randomly split the $N$ states into $\mathcal{S}_1$ and $\mathcal{S}_2$ such that $|\mathcal{S}_1| = \alpha N$ and $|\mathcal{S}_2| = (1 - \alpha)N$, where $\alpha$ is the proportion of states to be aliased. Then for every $s \in \mathcal{S}_1$, we randomly select an $\hat{s} \in \mathcal{S}_2$ and set $x_s \leftarrow x_{\hat{s}}$. Finally, we add Gaussian noise $\mathcal{N}(0, 0.1^2)$ to each element of $X$. We use $N = 100$ and $K = 30$ in our simulation and report the normalized representation error (NRE) as a function of $\gamma$. For a feature matrix $X$, the NRE is computed analytically as $\text{NRE}(\gamma) \doteq \frac{\min_w \|Xw - v_\gamma\|_2}{\|v_\gamma\|_2}$, where $v_\gamma$ is the analytically computed true value function of the MRP. We report the results in Figure 9, where each data point is averaged over $10^4$ randomly generated feature matrices ($X$) and reward functions. In this MRP, the average representation error becomes larger as $\gamma$ increases, which suggests that learning a good representation under a large $\gamma$ and state aliasing may be harder than with a smaller $\gamma$. We report the unnormalized representation error in Figure 17 in the appendix, where the trend is much clearer.

Overall, though we do not claim that there is a monotonic relationship between the discount factor and the difficulty of representation learning, our empirical study does suggest that representation learning is a key factor at play in the misuse of the discounting in actor-critic algorithms, beyond

---

[4]The trend that NRE decreases as $\alpha$ increases is merely an artifact from how we generate $v_\gamma$.

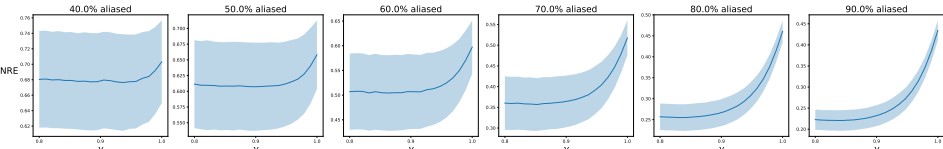

Figure 9: Normalized representation error as a function of the discount factor. Shaded regions indicate one standard derivation. [4]

the widely recognized bias-variance trade-off. In the appendix, we provide additional experiments involving distributional RL to further support the bias-variance-representation trade-off hypothesis, under the assumption that the benefits of distributional RL comes mainly from the improved representation learning (Bellemare et al., 2017; Munos, 2018; Petroski Such et al., 2019).

## 4 OPTIMIZING THE DISCOUNTED OBJECTIVE (SCENARIO 2)

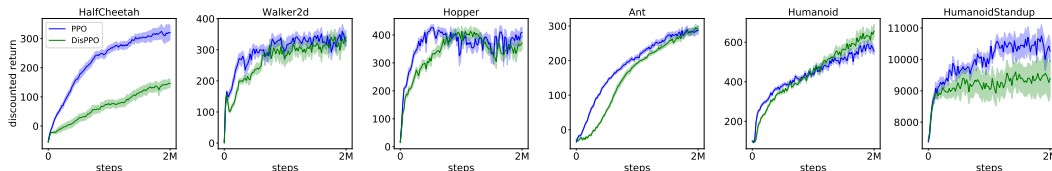

Figure 10: Comparison between PPO and DisPPO with $\gamma = 0.995$

When our goal is to optimize the *discounted* objective $J_{\gamma<1}(\pi)$, theoretically we should have the $\gamma_{\rm A}^t$ term in the actor update and use $\gamma_{\rm C} < 1$. Practitioners, however, usually ignore this $\gamma_{\rm A}^t$ (*i.e.*, set $\gamma_{\rm A} = 1$), introducing *bias*. Figure 10 shows that even if we use the *discounted* return as the performance metric, the biased implementation of PPO still outperforms the theoretically grounded implementation DisPPO in the domains we tested. Here PPO refers to the default PPO implementation where $\gamma_{\rm A} = 1, \gamma_{\rm C} = \gamma < 1$, and DisPPO (Alg. 6 in the appendix) adds the missing $\gamma_{\rm A}^t$ term in PPO by using $\gamma_{\rm A} = \gamma_{\rm C} = \gamma < 1$. We propose to interpret the empirical advantages of PPO over DisPPO with Hypothesis 2. For all experiments in this section, we use $\gamma_{\rm C} = \gamma < 1$.

**An auxiliary task perspective:** The biased policy update implementation of (2) ignoring $\gamma_{\rm A}^t$ can be decomposed into two parts as $\Delta_t = \gamma^t \Delta_t + (1 - \gamma^t)\Delta_t$, where $\Delta_t \doteq q_\pi^{\gamma_{\rm C}}(S_t, A_t)\nabla_\theta \log \pi(A_t|S_t)$. We propose to interpret the *difference term* between the biased implementation ($\Delta_t$) and the theoretically grounded implementation ($\gamma^t \Delta_t$), *i.e.*, the $(1-\gamma^t)q_\pi^{\gamma_{\rm C}}(S_t, A_t)\nabla_\theta \log \pi(A_t|S_t)$ term, as the gradient of an auxiliary objective with a dynamic weighting $1 - \gamma^t$. Let $J_{s,\mu}(\pi) \doteq \sum_a \pi(a|s)q_\mu^\gamma(s,a)$; we have $\nabla_\theta J_{s,\mu}(\pi)|_{\mu=\pi} = \mathbb{E}_{a\sim\pi(\cdot|s)}[q_\pi^\gamma(s,a)\nabla_\theta \log \pi(a|s)]$. This objective changes every time step (through $\mu$). Inspired by the decomposition, we augment PPO with this auxiliary task, yielding AuxPPO (Algorithm 7 and Figure 13 in the appendix). In AuxPPO, we have two policies $\pi$ and $\pi'$ parameterized by $\theta$ and $\theta'$ respectively. The two policies are two heads over the same neural network backbone, where $\pi$ is used for interaction with the environment and $\pi'$ is the policy for the auxiliary task. AuxPPO optimizes $\theta$ and $\theta'$ simultaneously by considering the following joint loss

$$L(\theta, \theta') \doteq \sum_{t=0}^\infty \gamma^t \min\left\{ \frac{\pi_\theta(A_t|S_t)}{\pi_{\theta_{\rm old}}(A_t|S_t)}{\rm Adv}_{\pi_{\theta_{\rm old}}}^{\gamma_{\rm C}}(S_t, A_t), {\rm clip}(\frac{\pi_\theta(A_t|S_t)}{\pi_{\theta_{\rm old}}(A_t|S_t)}){\rm Adv}_{\pi_{\theta_{\rm old}}}^{\gamma_{\rm C}}(S_t, A_t) \right\} +$$

$$\sum_{t=0}^\infty (1 - \gamma^t) \min\left\{ \frac{\pi_{\theta'}(A_t|S_t)}{\pi_{\theta_{\rm old}}(A_t|S_t)}{\rm Adv}_{\pi_{\theta_{\rm old}}}^{\gamma_{\rm C}}(S_t, A_t), {\rm clip}(\frac{\pi_{\theta'}(A_t|S_t)}{\pi_{\theta_{\rm old}}(A_t|S_t)}){\rm Adv}_{\pi_{\theta_{\rm old}}}^{\gamma_{\rm C}}(S_t, A_t) \right\},$$

where $S_t$ and $A_t$ are obtained by executing $\theta_{\rm old}$. We additionally synchronize $\theta'$ with $\theta$ periodically to avoid an off-policy learning issue.

**Flipped rewards:** Besides AuxPPO, we also design novel environments with flipped rewards to investigate Hypothesis 2. Recall we include the time step in the state, this allows us to simply create a new environment by defining a new reward function $r'(s, t) \doteq r(s)\mathbb{I}_{t\le t_0} - r(s)\mathbb{I}_{t>t_0}$, where $\mathbb{I}$ is the indicator function. During an episode, within the first $t_0$ steps, this new environment is the same as the original one. After $t_0$ steps, the sign of the reward is flipped. We select $t_0$ such that

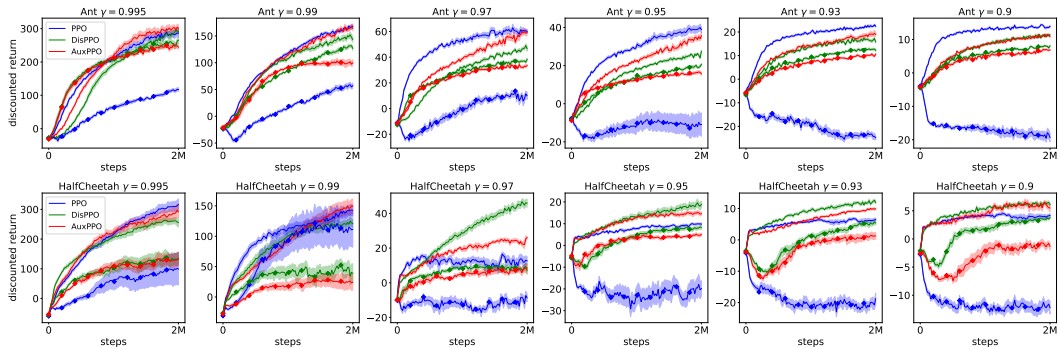

Figure 11: Curves without any marker are obtained in the original `Ant` / `HalfCheetah`. Diamond-marked curves are obtained in `Ant` / `HalfCheetah` with $r'$. [5]

$\gamma^{t_0}$ is sufficiently small, *e.g.*, we define $t_0 \doteq \min_t\{\gamma^t < 0.05\}$. With this criterion for selecting $t_0$, the later transitions (*i.e.*, transitions after $t_0$ steps) have little influence on the evaluation objective, the discounted return. Consequently, the later transitions affect the overall learning process mainly through representation learning. DisPPO rarely makes use of the later transitions due to the $\gamma_A^t$ term in the gradient update. AuxPPO makes use of the later transitions only through representation learning. PPO exploits the later transitions for representation learning and the later transitions also affect the control policy of PPO directly.

**Results:** When we consider the original environments, Figure 11 shows that in 8 out 12 tasks, PPO outperforms DisPPO, even if the performance metric is the *discounted* episodic return. In all those 8 tasks, by using the difference term as an auxiliary task, AuxPPO is able to improve upon DisPPO. In 6 out of those 8 tasks, AuxPPO is able to roughly match the performance of PPO at the end of training. For $\gamma \in \{0.93, 0.9\}$ in `Ant`, the improvement of AuxPPO is not clear and we conjecture that this is because the learning of the $\pi$-head (the control head) in AuxPPO is much slower than the learning of $\pi$ in PPO due to the $\gamma_C^t$ term. Overall, this suggests that the benefit of PPO over DisPPO comes mainly from representation learning.

When we consider the environments with flipped rewards, PPO is outperformed by DisPPO and AuxPPO by a large margin in 11 out of 12 tasks. The transitions after $t_0$ steps are not directly relevant when the performance metric is the discounted return. However, learning on those transitions may still improve representation learning provided that those transitions are similar to the earlier transitions, which is the case in the original environments. PPO and AuxPPO, therefore, outperform DisPPO. However, when those transitions are much different from the earlier transitions, which is the case in the environments with flipped rewards, learning to control on them directly becomes distracting. PPO, therefore, is outperformed by DisPPO. Different from PPO, AuxPPO does not learn to control on later transitions. Provided that the network has enough capacity, the control head $\pi_\theta$ in AuxPPO will not be affected much by the irrelevant transitions. The performance of AuxPPO is, therefore, similar to DisPPO.

To summarize, Figure 11 suggests that using $\gamma_A = 1$ is simply an *inductive bias* that *all transitions are equally important*. When this inductive bias is helpful for learning, $\gamma_A = 1$ implicitly implements auxiliary tasks thus improving representation learning and the overall performance. When this inductive bias is detrimental, however, $\gamma_A = 1$ can lead to significant performance drops. AuxPPO appears to be a safe choice that does not depend much on the correctness of this inductive bias.

## 5 RELATED WORK

The mismatch in actor-critic algorithm implementations has been previously studied. Thomas (2014) focuses on the natural policy gradient setting and shows that the biased implementation ignoring $\gamma_A^t$ can be interpreted as the gradient of the average reward objective under a strong assumption that the state distribution is independent of the policy. Nota & Thomas (2020) prove that

---

[5]See Section B.1 for more details about task selection.

without this strong assumption, the biased implementation is *not* the gradient of any *stationary* objective. This does not contradict our auxiliary task perspective as our objective $J_{s,\mu}(\pi)$ changes at every time step. Nota & Thomas (2020) further provide a counterexample showing that following the biased gradient can lead to a policy of poor performance w.r.t. both discounted and undiscounted objectives. Both Thomas (2014) and Nota & Thomas (2020), however, focus on *theoretical disadvantages* of the biased gradient and regard ignoring $\gamma_A^t$ as the source of the bias. We instead regard the introduction of $\gamma_C < 1$ in the critic as the source of the bias in the undiscounted setting and investigate its *empirical advantages*, which are more relevant to practitioners. Moreover, our representation learning perspective for investigating this mismatch is to our knowledge novel.

Although we propose the *bias-variance-representation* trade-off, we do not claim that is all that $\gamma$ affects. The discount factor also has many other effects (*e.g.*, Sutton (1995); Jiang et al. (2016); Laroche et al. (2017); Laroche & van Seijen (2018); Lehnert et al. (2018); Fedus et al. (2019); Van Seijen et al. (2019); Amit et al. (2020)), which we leave for future work. In Scenario 1, using $\gamma_C < 1$ helps reduce the variance. Variance reduction in RL itself is an active research area (see, e.g., Papini et al. (2018); Xu et al. (2019); Yuan et al. (2020)). Investigating those variance reduction techniques with $\gamma_C = 1$ is a possibility for future work. Recently, Bengio et al. (2020) study the effect of the bootstrapping parameter $\lambda$ in TD($\lambda$) in generalization. Our work studies the effect of the discount factor $\gamma$ in representation learning in the context of the misuse of the discounting in actor-critic algorithms, sharing a similar spirit of Bengio et al. (2020).

## 6 CONCLUSION

In this paper, we investigate the longstanding mismatch between theorists and practitioners in actor-critic algorithms from a representation learning perspective. Although the theoretical understanding of policy gradient algorithms have recently been significantly advanced (Agarwal et al., 2019; Wu et al., 2020), this mismatch has drawn little attention. We hope our empirical study can help practitioners understand actor-critic algorithms better and therefore design more efficient actor-critic algorithms in the setting of deep RL, where representation learning emerges as a major consideration. We hope our empirical study can draw more attention to the mismatch, which could enable the community to finally close this longstanding gap.

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

# A   PROOF OF LEMMA 2

*Proof.* The proof is based on Appendix B in Schulman et al. (2015a), where perturbation theory is used to prove the performance improvement bound (Lemma 1). To simplify notation, we use a vector and a function interchangeably, *i.e.*, we also use $r$ and $\mu_0$ to denote the reward vector and the initial distribution vector. $J(\pi)$ and $d_\pi(s)$ are shorthand for $J_\gamma(\pi)$ and $d_\pi^\gamma(s)$ with $\gamma = 1$. All vectors are *column* vectors.

Let $\mathcal{S}^+$ be the set of states excluding $s^\infty$, *i.e.*, $\mathcal{S}^+ \doteq \mathcal{S}/\{s^\infty\}$, we define $P_\pi \in \mathbb{R}^{|\mathcal{S}^+| \times |\mathcal{S}^+|}$ such that $P_\pi(s, s') \doteq \sum_a \pi(a|s)p(s'|s, a)$. Let $G \doteq \sum_{t=0}^\infty P_\pi^t$. According to standard Markov chain theories, $G(s, s')$ is the expected number of times that $s'$ is visited before $s^\infty$ is hit given $S_0 = s$. $T_{\max} < \infty$ implies that $G$ is well-defined and we have $G = (I - P_\pi)^{-1}$. Moreover, $T_{\max} < \infty$ also implies $\forall s, \sum_{s'} G(s, s') \leq T_{\max}$, *i.e.*, $||G||_\infty \leq T_{\max}$. We have $J(\pi) = \mu_0^\top Gr$.

Let $G' \doteq (I - P_{\pi'})^{-1}$, we have

$$J(\pi') - J(\pi) = \mu_0^\top (G' - G)r.$$

Let $\Delta \doteq P_{\pi'} - P_\pi$, we have

$$G'^{-1} - G^{-1} = -\Delta,$$

Left multiply by $G'$ and right multiply by $G$,

$$
\begin{aligned}
G - G' &= -G'\Delta G, \\
G' &= G + G'\Delta G \quad \text{(Expanding } G' \text{ in RHS recursively)} \\
&= G + G\Delta G + G'\Delta G\Delta G.
\end{aligned}
$$

So we have

$$J(\pi') - J(\pi) = \mu_0^\top G\Delta Gr + \mu_0^\top G'\Delta G\Delta Gr.$$

It is easy to see $\mu_0^\top G = d_\pi^\top$ and $Gr = v_\pi$. So

$$
\begin{aligned}
\mu_0^\top G\Delta Gr &= d_\pi^\top \Delta v_\pi \\
&= \sum_s d_\pi(s) \sum_{s'} \Big( \sum_a \pi'(a|s)p(s'|s, a) - \sum_a \pi(a|s)p(s'|s, a) \Big) v_\pi(s') \\
&= \sum_s d_\pi(s) \sum_a (\pi'(a|s) - \pi(a|s)) \sum_{s'} p(s'|s, a)v_\pi(s') \\
&= \sum_s d_\pi(s) \sum_a (\pi'(a|s) - \pi(a|s)) \Big( r(s) + \sum_{s'} p(s'|s, a)v_\pi(s') - v_\pi(s) \Big)
\end{aligned}
$$

$$(\textstyle\sum_a (\pi'(a|s) - \pi(a|s))f(s) = 0 \text{ holds for any } f \text{ that dependes only on } s)$$

$$= \sum_s d_\pi(s) \sum_a \pi'(a|s)\text{Adv}_\pi(s, a).$$

$$(\textstyle\sum_a \pi(a|s)\text{Adv}_\pi(s, a) = 0 \text{ by Bellman equation})$$

We now bound $\mu_0^\top G'\Delta G\Delta Gr$. First,

$$
\begin{aligned}
|(\Delta Gr)(s)| &= |\sum_{s'} \Big( \sum_a \pi'(a|s) - \pi(a|s) \Big) p(s'|s, a)v_\pi(s')| \\
&= |\sum_a \Big( \pi'(a|s) - \pi(a|s) \Big) \Big( r(s) + \sum_{s'} p(s'|s, a)v_\pi(s') - v_\pi(s) \Big)| \\
&= |\sum_a \Big( \pi'(a|s) - \pi(a|s) \Big) \text{Adv}_\pi(s, a)| \\
&\leq 2 \max_s \text{D}_{TV}(\pi'(\cdot|s), \pi(\cdot|s)) \max_{s,a} |\text{Adv}_\pi(s, a)|,
\end{aligned}
$$

where $D_{TV}$ is the total variation distance. So

$$||\Delta Gr||_\infty \leq 2\max_s D_{TV}(\pi'(\cdot|s), \pi(\cdot|s))\max_{s,a}|\text{Adv}_\pi(s,a)|.$$

Moreover, for any vector $x$,

$$|(\Delta x)(s)| \leq 2\max_s D_{TV}(\pi'(\cdot|s), \pi(\cdot|s))||x||_\infty,$$
$$||\Delta x||_\infty \leq 2\max_s D_{TV}(\pi'(\cdot|s), \pi(\cdot|s))||x||_\infty.$$

So

$$||\Delta||_\infty \leq 2\max_s D_{TV}(\pi'(\cdot|s), \pi(\cdot|s)),$$
$$|\mu_0^\top G'\Delta G\Delta Gr| \leq ||\mu_0^\top||_1||G'||_\infty||\Delta||_\infty||G||_\infty||\Delta Gr||_\infty$$
$$\leq 4T_{\max}^2\max_s D_{TV}^2(\pi'(\cdot|s), \pi(\cdot|s))\max_{s,a}|\text{Adv}_\pi(s,a)|$$
$$\leq 4T_{\max}^2\max_s D_{KL}(\pi(\cdot|s)||\pi'(\cdot|s))\max_{s,a}|\text{Adv}_\pi(s,a)|,$$

which completes the proof. $\square$

Note this perturbation-based proof of Lemma 2 holds only for $r : \mathcal{S} \to \mathbb{R}$. For $r : \mathcal{S} \times \mathcal{A} \to \mathbb{R}$, we can turn to the coupling-based proof as Schulman et al. (2015a), which, however, complicates the presentation and deviates from the main purpose of this paper. We, therefore, leave it for future work.

## B    EXPERIMENT DETAILS

### B.1    METHODOLOGY

We use `HalfCheetah`, `Walker`, `Hopper`, `Ant`, `Humanoid`, and `HumanoidStandup` as our benchmarks. We exclude other tasks as we find PPO plateaus quickly there. The tasks we consider have a hard time limit of 1000. Following Pardo et al. (2018), we add time step information into the state, *i.e.*, there is an additional scalar $t/1000$ in the observation vector. Following Achiam (2018), we estimate the KL divergence between the current policy $\theta$ and the sampling policy $\theta_{\text{old}}$ when optimizing the loss (3). When the estimated KL divergence is greater than a threshold, we stop updating the actor and update only the critic with current data. We use Adam (Kingma & Ba, 2014) as the optimizer and perform grid search for the initial learning rates of Adam optimizers. Let $\alpha_A$ and $\alpha_C \doteq \beta\alpha_A$ be the learning rates for the actor and critic respectively. For each algorithmic configuration (*i.e.*, a curve in a figure), we tune $\alpha_A \in \{0.125, 0.25, 0.5, 1, 2\} \times 3 \cdot 10^{-4}$ and $\beta \in \{1, 3\}$ with grid search in `Ant` with 3 independent runs maximizing the average return of the last 100 training episodes. In particular, $\alpha_A = 3 \cdot 10^{-4}$ and $\beta = 3$ is roughly the default learning rates for the PPO implementation in Achiam (2018). We then run this algorithmic configuration with the best $\alpha_A$ and $\alpha_C$ in all tasks. Overall, we find after removing GAE, smaller learning rates are preferred. When we use FHTD, we additionally consider $H \in \{16, 32, 64, 128, 256, 512, 1024\}$ in the grid search. When we use C51, we additionally consider $V_{\max} \in \{20, 40, 80, 160, 320, 640, 1280, 2560, 5120, 10240, 81920, 163840, 327680\}$ in the grid search. We use PPO-TD with $\gamma_C = 0.99$ as an example to study how the best hyperparameter configuration in `Ant` transfers to other games. As shown in Figure 12, the best learning rates of `Ant` ($\alpha_A = 3 \cdot 10^{-4}$ and $\beta = 3$) yields reasonably good performance in all the other games except `Humanoid`. In the paper, we do not draw a conclusion from a single task. So an outlier is unlikely to affect the overall conclusion.

In the discounted setting, we consider only `Ant`, `HalfCheetah` and their variants. For `Walker2d`, `Hopper`, and `Humanoid`, we find the average episode length of all algorithms are smaller than $t_0$, *i.e.*, the flipped reward rarely takes effects. For `HumanoidStandup`, the scale of the reward is too large. To summarize, other four environments are not well-suited for the purpose of our empirical study. Moreover, in the discounted setting, we performed the grid search of the learning rates for both `Ant` and `HalfCheetah`.

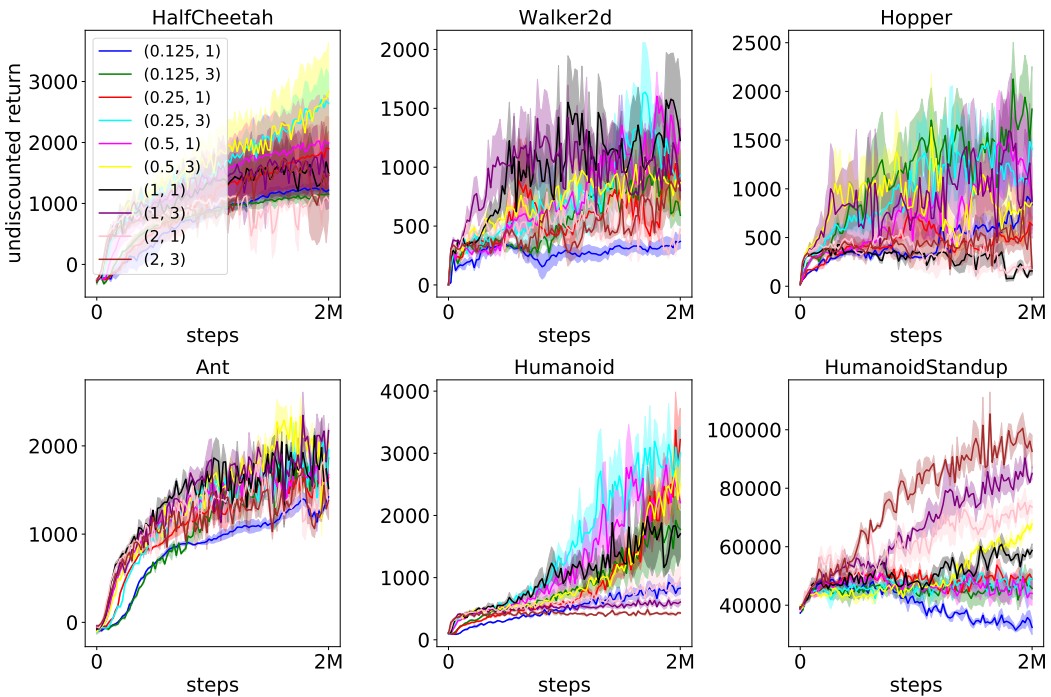

Figure 12: PPO-TD ($\gamma_{\mathrm{C}} = 0.99$) with different learning rates. A curve labeled with $(x, \beta)$ corresponds to an initial learning rate for the actor and critic of $\alpha_{\mathrm{A}} = x \times 3 \cdot 10^{-4}$ and $\alpha_{\mathrm{C}} = \beta \alpha_{\mathrm{A}}$ respectively. The best learning rates for `Ant` ($\alpha_{\mathrm{A}} = 3 \cdot 10^{-4}$ and $\beta = 3$) yields reasonably good performance in all the other games except `Humanoid`.

## B.2 ALGORITHM DETAILS

The pseudocode of all implemented algorithms are provide in Algorithms 1 - 7 with their architectures illustrated in Figure 13. For hyperparameters that are not included in the grid search, we use the same value as Dhariwal et al. (2017); Achiam (2018). In particular, for the rollout length, we set $K = 2048$. For the optimization epochs, we set $K_{opt} = 320$. For the minibatch size, we set $B = 64$. For the maximum KL divergence, we set $KL_{target} = 0.01$. We clip $\frac{\pi_\theta(a|s)}{\pi_{\theta_{old}}(a|s)}$ into $[-0.2, 0.2]$. We use $N_s = 51$ supports for PPO-C51.

We use two-hidden-layer neural networks for function approximation. Each hidden layer has 64 hidden units and a `tanh` activation function. The output layer of the actor network has a `tanh` activation function and is interpreted as the mean of an isotropic Gaussian distribution, whose standard derivation is a global state-independent variable as suggested by Schulman et al. (2015a).

---

**Algorithm 1:** PPO

---

**Input:**

$\theta, \psi$: parameters of $\pi, \hat{v}$

$\alpha_A, \alpha_C$: Initial learning rates of the Adam optimizers for $\theta, \psi$

$K, K_{opt}, B$: rollout length, number of optimization epochs, and minibatch size

$KL_{target}$: maximum KL divergence threshold

$S_0 \sim \mu_0$

**while** *True* **do**

    Initialize a buffer $M$

    $\theta_{old} \leftarrow \theta$

    **for** $i = 0, \ldots, K - 1$ **do**

        $A_i \sim \pi_{\theta_{old}}(\cdot|S_i)$

        Execute $A_i$, get $R_{i+1}, S_{i+1}$

        **if** $S_{i+1}$ *is a terminal state* **then**

            $m_i \leftarrow 0, S_{i+1} \sim \mu_0$

        **else**

            $m_i \leftarrow 1$

        **end**

    **end**

    $G_K \leftarrow \hat{v}(S_K)$

    **for** $i = K - 1, \ldots, 0$ **do**

        $G_i \leftarrow R_{i+1} + \gamma_C m_i G_{i+1}$

        $\text{Adv}_i \leftarrow R_{i+1} + \gamma_C m_i \hat{v}_\psi(S_{i+1}) - \hat{v}_\psi(S_i)$

        Store $(S_i, A_i, G_i, \text{Adv}_i)$ in $M$

    **end**

    Normalize $\text{Adv}_i$ in $M$ as $\text{Adv}_i \leftarrow \frac{\text{Adv}_i - \text{mean}(\{\text{Adv}_i\})}{\text{std}(\{\text{Adv}_i\})}$

    **for** $o = 1, \ldots, K_{opt}$ **do**

        Sample a minibatch $\{(S_i, A_i, G_i, \text{Adv}_i)\}_{i=1,\ldots,B}$ from $M$

        $L(\psi) \leftarrow \frac{1}{2B} \sum_{i=1}^{B} (\hat{v}_\psi(S_i) - G_i)^2$ `/* No gradient through` $G_i$ `*/`

        $L(\theta) \leftarrow \frac{1}{B} \sum_{i=1}^{B} \min\{\frac{\pi_\theta(A_i|S_i)}{\pi_{\theta_{old}}(A_i|S_i)} \text{Adv}_i, \text{clip}(\frac{\pi_\theta(A_i|S_i)}{\pi_{\theta_{old}}(A_i|S_i)}) \text{Adv}_i\}$

        Perform one gradient update to $\psi$ minimizing $L(\psi)$ with Adam

        **if** $\frac{1}{B} \sum_{i=1}^{B} \log \pi_{\theta_{old}}(A_i|S_i) - \log \pi_\theta(A_i|S_i) < KL_{target}$ **then**

            Perform one gradient update to $\theta$ maximizing $L(\theta)$ with Adam

        **end**

    **end**

**end**

---

---

**Algorithm 2:** PPO-TD

---

**Input:**

$\theta, \psi$: parameters of $\pi, \hat{v}$

$\alpha_A, \alpha_C$: Initial learning rates of the Adam optimizers for $\theta, \psi$

$K, K_{opt}, B$: rollout length, number of optimization epochs, and minibatch size

$KL_{target}$: maximum KL divergence threshold

$S_0 \sim \mu_0$
**while** *True* **do**
  Initialize a buffer $M$
  $\theta_{old} \leftarrow \theta$
  **for** $i = 0, \ldots, K - 1$ **do**
    $A_i \sim \pi_{\theta_{old}}(\cdot | S_i)$
    Execute $A_i$, get $R_{i+1}, S_{i+1}$
    **if** $S_{i+1}$ *is a terminal state* **then**
      $m_i \leftarrow 0, S_{i+1} \sim \mu_0$
    **else**
      $m_i \leftarrow 1$
    **end**
  **end**
  **for** $i = K - 1, \ldots, 0$ **do**
    $\text{Adv}_i \leftarrow R_{i+1} + \gamma_{\text{C}} m_i \hat{v}_\psi(S_{i+1}) - \hat{v}_\psi(S_i)$
    $S_i' \leftarrow S_{i+1}, r_i \leftarrow R_{i+1}$
    Store $(S_i, A_i, m_i, r_i, S_i', \text{Adv}_i)$ in $M$
  **end**
  Normalize $\text{Adv}_i$ in $M$ as $\text{Adv}_i \leftarrow \frac{\text{Adv}_i - \text{mean}(\{\text{Adv}_i\})}{\text{std}(\{\text{Adv}_i\})}$
  **for** $o = 1, \ldots, K_{opt}$ **do**
    Sample a minibatch $\{(S_i, A_i, m_i, r_i, S_i', \text{Adv}_i)\}_{i=1,\ldots,B}$ from $M$
    $y_i \leftarrow r_i + \gamma_{\text{C}} m_i \hat{v}_\psi(S_i')$
    $L(\psi) \leftarrow \frac{1}{2B} \sum_{i=1}^{B} (\hat{v}_\psi(S_i) - y_i)^2$ /* No gradient through $y_i$                   */
    $L(\theta) \leftarrow \frac{1}{B} \sum_{i=1}^{B} \min\{\frac{\pi_\theta(A_i|S_i)}{\pi_{\theta_{old}}(A_i|S_i)} \text{Adv}_i, \text{clip}(\frac{\pi_\theta(A_i|S_i)}{\pi_{\theta_{old}}(A_i|S_i)}) \text{Adv}_i\}$
    Perform one gradient update to $\psi$ minimizing $L(\psi)$ with Adam
    **if** $\frac{1}{B} \sum_{i=1}^{B} \log \pi_{\theta_{old}}(A_i|S_i) - \log \pi_\theta(A_i|S_i) < KL_{target}$ **then**
      Perform one gradient update to $\theta$ maximizing $L(\theta)$ with Adam
    **end**
  **end**
**end**

---

**Algorithm 3:** PPO-TD-Ex

---

**Input:**

$\theta, \psi$: parameters of $\pi, \hat{v}$

$\alpha_A, \alpha_C$: Initial learning rates of the Adam optimizers for $\theta, \psi$

$K, K_{opt}, B$: rollout length, number of optimization epochs, and minibatch size

$KL_{target}$: maximum KL divergence threshold

$N$: number of extra transitions

$p, r$: transition kernel and reward function of the oracle

$S_0 \sim \mu_0$

**while** *True* **do**

    Initialize a buffer $M$

    $\theta_{old} \leftarrow \theta$

    **for** $i = 0, \ldots, K-1$ **do**

        **for** $j = 0, \ldots, N$ **do**

            $A_i^j \sim \pi_{\theta_{old}}(\cdot|S_i), R_{i+1}^j \leftarrow r(S_i, A_i^j), S_{i+1}^j \sim p(\cdot|S_i, A_i^j)$

            **if** $S_{i+1}^j$ *is a terminal state* **then**

                $m_i^j \leftarrow 0, S_{i+1}^j \sim \mu_0$

            **else**

                $m_i^j \leftarrow 1$

            **end**

        **end**

        $S_{i+1} \leftarrow S_{i+1}^0$

    **end**

    **for** $i = K-1, \ldots, 0$ **do**

        $\text{Adv}_i \leftarrow R_{i+1}^0 + \gamma_C m_i^0 \hat{v}_\psi(S_{i+1}^0) - \hat{v}_\psi(S_i^0)$

        **for** $j = 0, \ldots, N$ **do**

            $S_i'^j \leftarrow S_{i+1}^j$

        **end**

        Store $(\{S_i^j, A_i^j, m_i^j, r_i^j, S_i'^j\}_{j=0,\ldots,N}, \text{Adv}_i)$ in $M$

    **end**

    Normalize $\text{Adv}_i$ in $M$ as $\text{Adv}_i \leftarrow \frac{\text{Adv}_i - \text{mean}(\{\text{Adv}_i\})}{\text{std}(\{\text{Adv}_i\})}$

    **for** $o = 1, \ldots, K_{opt}$ **do**

        Sample a minibatch $\{(\{S_i^j, A_i^j, m_i^j, r_i^j, S_i'^j\}_{j=0,\ldots,N}, \text{Adv}_i)\}_{i=1,\ldots,B}$ from $M$

        $y_i \leftarrow \frac{1}{N+1} \sum_{j=0}^N r_i^j + \gamma_C m_i^j \hat{v}_\psi(S_i'^j)$

        $L(\psi) \leftarrow \frac{1}{2B} \sum_{i=1}^B (\hat{v}_\psi(S_i^0) - y_i)^2$ /* No gradient through $y_i$         */

        $L(\theta) \leftarrow \frac{1}{B} \sum_{i=1}^B \min\{\frac{\pi_\theta(A_i^0|S_i^0)}{\pi_{\theta_{old}}(A_i^0|S_i^0)} \text{Adv}_i, \text{clip}(\frac{\pi_\theta(A_i^0|S_i^0)}{\pi_{\theta_{old}}(A_i^0|S_i^0)}) \text{Adv}_i\}$

        Perform one gradient update to $\psi$ minimizing $L(\psi)$ with Adam

        **if** $\frac{1}{B} \sum_{i=1}^B \log \pi_{\theta_{old}}(A_i^0|S_i^0) - \log \pi_\theta(A_i^0|S_i^0) < KL_{target}$ **then**

            Perform one gradient update to $\theta$ maximizing $L(\theta)$ with Adam

        **end**

    **end**

**end**

---

---

**Algorithm 4:** PPO-FHTD

---

**Input:**
$\theta, \psi$: parameters of $\pi, \{\hat{v}^j\}_{j=1,\dots,H}$
$\alpha_A, \alpha_C$: Initial learning rates of the Adam optimizers for $\theta, \psi$
$K, K_{opt}, B$: rollout length, number of optimization epochs, and minibatch size
$KL_{target}$: maximum KL divergence threshold

$S_0 \sim \mu_0$
**while** *True* **do**
    Initialize a buffer $M$
    $\theta_{old} \leftarrow \theta$
    **for** $i = 0, \dots, K - 1$ **do**
        $A_i \sim \pi_{\theta_{old}}(\cdot|S_i)$
        Execute $A_i$, get $R_{i+1}, S_{i+1}$
        **if** $S_{i+1}$ *is a terminal state* **then**
            $m_i \leftarrow 0, S_{i+1} \sim \mu_0$
        **else**
            $m_i \leftarrow 1$
        **end**
    **end**
    **for** $i = K - 1, \dots, 0$ **do**
        $\text{Adv}_i \leftarrow R_{i+1} + m_i \hat{v}_\psi^H(S_{i+1}) - \hat{v}_\psi^H(S_i)$
        $S_i' \leftarrow S_{i+1}, r_i \leftarrow R_{i+1}$
        Store $(S_i, A_i, m_i, r_i, S_i', \text{Adv}_i)$ in $M$
    **end**
    Normalize $\text{Adv}_i$ in $M$ as $\text{Adv}_i \leftarrow \frac{\text{Adv}_i - \text{mean}(\{\text{Adv}_i\})}{\text{std}(\{\text{Adv}_i\})}$
    **for** $o = 1, \dots, K_{opt}$ **do**
        Sample a minibatch $\{(S_i, A_i, m_i, r_i, S_i', \text{Adv}_i)\}_{i=1,\dots,B}$ from $M$
        **for** $j = 1, \dots, H$ **do**
            $y_i^j \leftarrow r_i + m_i \hat{v}_\psi^{j-1}(S_i'))$    /*  $\hat{v}^0(S_i') \equiv 0$            */
        **end**
        $L(\psi) \leftarrow \frac{1}{2B} \sum_{i=1}^B \sum_{j=1}^H (\hat{v}_\psi^j(S_i) - y_i^j)^2$ /* No gradient through $y_i^j$     */
        $L(\theta) \leftarrow \frac{1}{B} \sum_{i=1}^B \min\{\frac{\pi_\theta(A_i|S_i)}{\pi_{\theta_{old}}(A_i|S_i)} \text{Adv}_i, \text{clip}(\frac{\pi_\theta(A_i|S_i)}{\pi_{\theta_{old}}(A_i|S_i)}) \text{Adv}_i\}$
        Perform one gradient update to $\psi$ minimizing $L(\psi)$ with Adam
        **if** $\frac{1}{B} \sum_{i=1}^B \log \pi_{\theta_{old}}(A_i|S_i) - \log \pi_\theta(A_i|S_i) < KL_{target}$ **then**
            Perform one gradient update to $\theta$ maximizing $L(\theta)$ with Adam
        **end**
    **end**
**end**

---

---

**Algorithm 5:** PPO-C51

---

**Input:**

$\theta, \psi$: parameters of $\pi, \{\hat{v}^j\}_{j=1,\ldots,N_s}$ with $N_s$ being the number of supports and $\hat{v}^j$ being the probability of each support

$\alpha_A, \alpha_C$: Initial learning rates of the Adam optimizers for $\theta, \psi$

$K, K_{opt}, B$: rollout length, number of optimization epochs, and minibatch size

$KL_{target}$: maximum KL divergence threshold

$\Delta_z \doteq \frac{2V_{\max}}{N_s-1}, \{z_j \doteq -V_{\max} + (j-1)\Delta_z : j = 1, \ldots, N_s\}$ // Define the supports

$S_0 \sim \mu_0$

**while** *True* **do**

    Initialize a buffer $M$

    $\theta_{old} \leftarrow \theta$

    **for** $i = 0, \ldots, K - 1$ **do**

        $A_i \sim \pi_{\theta_{old}}(\cdot|S_i)$

        Execute $A_i$, get $R_{i+1}, S_{i+1}$

        **if** $S_{i+1}$ *is a terminal state* **then**

            $m_i \leftarrow 0, S_{i+1} \sim \mu_0$

        **else**

            $m_i \leftarrow 1$

        **end**

    **end**

    **for** $i = K - 1, \ldots, 0$ **do**

        $\text{Adv}_i \leftarrow R_{i+1} + m_i \gamma_C \sum_{j=1}^{N_s} \hat{v}_\psi^j(S_{i+1}) z_j - \sum_{j=1}^{N_s} \hat{v}_\psi^j(S_i) z_j$

        $S_i' \leftarrow S_{i+1}, r_i \leftarrow R_{i+1}$

        Store $(S_i, A_i, m_i, r_i, S_i', \text{Adv}_i)$ in $M$

    **end**

    Normalize $\text{Adv}_i$ in $M$ as $\text{Adv}_i \leftarrow \frac{\text{Adv}_i - \text{mean}(\{\text{Adv}_i\})}{\text{std}(\{\text{Adv}_i\})}$

    **for** $o = 1, \ldots, K_{opt}$ **do**

        Sample a minibatch $\{(S_i, A_i, m_i, r_i, S_i', \text{Adv}_i)\}_{i=1,\ldots,B}$ from $M$

        **for** $i = 1, \ldots, B$ **do**

            **for** $j = 1, \ldots, N_s$ **do**

                $z_j^i \leftarrow r_i + m_i \gamma_C z_j$

            **end**

        **end**

        **for** $j = 1, \ldots, N_s$ **do**

            $y_j^i \leftarrow \sum_{k=1}^{N_s}[1 - \frac{|[z_j^i]_{-V_{\max}}^{V_{\max}} - z_j|}{\Delta_z}]_0^1 \hat{v}_\psi^k(S_i')$ /* $[x]_l^u \doteq \min(\max(x, l), u)$ */

        **end**

        $L(\psi) \leftarrow \frac{1}{B} \sum_{i=1}^{B} \sum_{j=1}^{N_s} -y_j^i \log \hat{v}_\psi^j(S_i)$ /* No gradient through $y_j^i$ */

        $L(\theta) \leftarrow \frac{1}{B} \sum_{i=1}^{B} \min\{\frac{\pi_\theta(A_i|S_i)}{\pi_{\theta_{old}}(A_i|S_i)}\text{Adv}_i, \text{clip}(\frac{\pi_\theta(A_i|S_i)}{\pi_{\theta_{old}}(A_i|S_i)})\text{Adv}_i\}$

        Perform one gradient update to $\psi$ minimizing $L(\psi)$ with Adam

        **if** $\frac{1}{B} \sum_{i=1}^{B} \log \pi_{\theta_{old}}(A_i|S_i) - \log \pi_\theta(A_i|S_i) < KL_{target}$ **then**

            Perform one gradient update to $\theta$ maximizing $L(\theta)$ with Adam

        **end**

    **end**

**end**

---

---

**Algorithm 6:** DisPPO

---

**Input:**
$\theta, \psi$: parameters of $\pi, \hat{v}$
$\alpha_A, \alpha_C$: Initial learning rates of the Adam optimizers for $\theta, \psi$
$K, K_{opt}, B$: rollout length, number of optimization epochs, and minibatch size
$KL_{target}$: maximum KL divergence threshold

$S_0 \sim \mu_0, t \leftarrow 0$
**while** *True* **do**

    Initialize a buffer $M$
    $\theta_{old} \leftarrow \theta$
    **for** $i = 0, \ldots, K - 1$ **do**
        $A_i \sim \pi_{\theta_{old}}(\cdot | S_i), t_i \leftarrow t$
        Execute $A_i$, get $R_{i+1}, S_{i+1}$
        **if** $S_{i+1}$ *is a terminal state* **then**
            $m_i \leftarrow 0, S_{i+1} \sim \mu_0, t \leftarrow 0$
        **else**
            $m_i \leftarrow 1, t \leftarrow t + 1$
        **end**
    **end**
    $G_K \leftarrow \hat{v}(S_K)$
    **for** $i = K - 1, \ldots, 0$ **do**
        $G_i \leftarrow R_{i+1} + \gamma_C m_i G_{i+1}$
        $\text{Adv}_i \leftarrow R_{i+1} + \gamma_C m_i \hat{v}_\psi(S_{i+1}) - \hat{v}_\psi(S_i)$
        Store $(S_i, A_i, G_i, \text{Adv}_i, t_i)$ in $M$
    **end**

    Normalize $\text{Adv}_i$ in $M$ as $\text{Adv}_i \leftarrow \frac{\text{Adv}_i - \text{mean}(\{\text{Adv}_i\})}{\text{std}(\{\text{Adv}_i\})}$
    **for** $o = 1, \ldots, K_{opt}$ **do**
        Sample a minibatch $\{(S_i, A_i, G_i, \text{Adv}_i, t_i)\}_{i=1,\ldots,B}$ from $M$
        $L(\psi) \leftarrow \frac{1}{2B} \sum_{i=1}^{B} (\hat{v}_\psi(S_i) - G_i)^2$ /* No gradient through $G_i$ */
        $L(\theta) \leftarrow \frac{1}{B} \sum_{i=1}^{B} \gamma_A^{t_i} \min\{\frac{\pi_\theta(A_i|S_i)}{\pi_{\theta_{old}}(A_i|S_i)} \text{Adv}_i, \text{clip}(\frac{\pi_\theta(A_i|S_i)}{\pi_{\theta_{old}}(A_i|S_i)}) \text{Adv}_i\}$
        Perform one gradient update to $\psi$ minimizing $L(\psi)$ with Adam
        **if** $\frac{1}{B} \sum_{i=1}^{B} \log \pi_{\theta_{old}}(A_i|S_i) - \log \pi_\theta(A_i|S_i) < KL_{target}$ **then**
            Perform one gradient update to $\theta$ maximizing $L(\theta)$ with Adam
        **end**
    **end**
**end**

---

---

**Algorithm 7:** AuxPPO

---

**Input:**
$\theta, \theta', \psi$: parameters of $\pi, \pi', \hat{v}$
$\alpha_A, \alpha_C$: Initial learning rates of the Adam optimizers for $\theta, \psi$
$K, K_{opt}, B$: rollout length, number of optimization epochs, and minibatch size
$KL_{target}$: maximum KL divergence threshold

$S_0 \sim \mu_0, t \leftarrow 0$
**while** *True* **do**
    Initialize a buffer $M$
    $\theta_{old} \leftarrow \theta, \textcolor{red}{\theta' \leftarrow \theta}$
    **for** $i = 0, \ldots, K - 1$ **do**
        $A_i \sim \pi_{\theta_{old}}(\cdot|S_i), t_i \leftarrow t$
        Execute $A_i$, get $R_{i+1}, S_{i+1}$
        **if** $S_{i+1}$ *is a terminal state* **then**
            $m_i \leftarrow 0, S_{i+1} \sim \mu_0, t \leftarrow 0$
        **else**
            $m_i \leftarrow 1, t \leftarrow t + 1$
        **end**
    **end**
    $G_K \leftarrow \hat{v}(S_K)$
    **for** $i = K - 1, \ldots, 0$ **do**
        $G_i \leftarrow R_{i+1} + \gamma_C m_i G_{i+1}$
        $\text{Adv}_i \leftarrow R_{i+1} + \gamma_C m_i \hat{v}_\psi(S_{i+1}) - \hat{v}_\psi(S_i)$
        Store $(S_i, A_i, G_i, \text{Adv}_i, t_i)$ in $M$
    **end**
    Normalize $\text{Adv}_i$ in $M$ as $\text{Adv}_i \leftarrow \frac{\text{Adv}_i - \text{mean}(\{\text{Adv}_i\})}{\text{std}(\{\text{Adv}_i\})}$
    **for** $o = 1, \ldots, K_{opt}$ **do**
        Sample a minibatch $\{(S_i, A_i, G_i, \text{Adv}_i, t_i)\}_{i=1,\ldots,B}$ from $M$
        $L(\psi) \leftarrow \frac{1}{2B} \sum_{i=1}^{B} (\hat{v}_\psi(S_i) - G_i)^2$ /* No gradient through $G_i$          */

$$L(\theta, \theta') \leftarrow \frac{1}{B} \sum_{i=1}^{B} \textcolor{red}{\gamma_C^{t_i}} \min\{\frac{\pi_\theta(A_i|S_i)}{\pi_{\theta_{old}}(A_i|S_i)}\text{Adv}_i, \text{clip}(\frac{\pi_\theta(A_i|S_i)}{\pi_{\theta_{old}}(A_i|S_i)})\text{Adv}_i\} +$$
$$\textcolor{red}{(1 - \gamma_C^{t_i})} \min\{\frac{\pi_{\theta'}(A_i|S_i)}{\pi_{\theta_{old}}(A_i|S_i)}\text{Adv}_i, \text{clip}(\frac{\pi_{\theta'}(A_i|S_i)}{\pi_{\theta_{old}}(A_i|S_i)})\text{Adv}_i\}$$

        Perform one gradient update to $\psi$ minimizing $L(\psi)$ with Adam
        **if** $\frac{1}{B} \sum_{i=1}^{B} \log \pi_{\theta_{old}}(A_i|S_i) - \log \pi_\theta(A_i|S_i) < KL_{target}$ **then**
            Perform one gradient update to $\theta, \theta'$ maximizing $L(\theta, \theta')$ with Adam
        **end**
    **end**
**end**

---

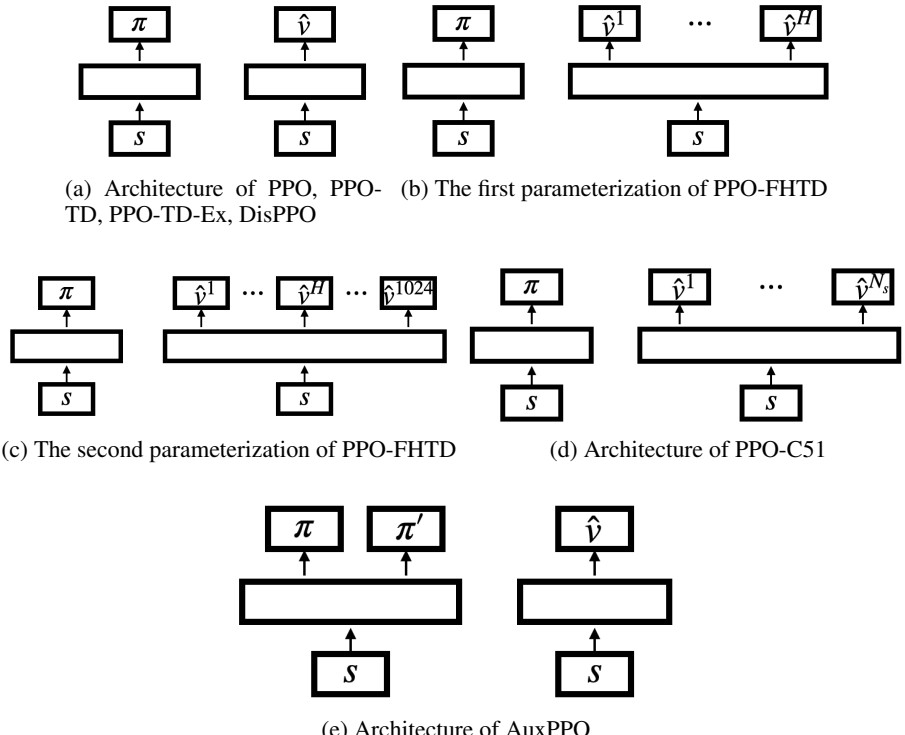

(a) Architecture of PPO, PPO-TD, PPO-TD-Ex, DisPPO

(b) The first parameterization of PPO-FHTD

(c) The second parameterization of PPO-FHTD

(d) Architecture of PPO-C51

(e) Architecture of AuxPPO

Figure 13: Architectures of the algorithms

## C  ADDITIONAL EXPERIMENTAL RESULTS

### C.1  DISTRIBUTIONAL RL

Hypothesis 1 and the previous empirical study suggest that representation learning may be the main bottleneck of PPO-TD ($\gamma_{\mathrm{C}} = 1$). To further support this, we benchmark PPO-C51 ($\gamma_{\mathrm{C}} = 1$) (Algorithm 5 in the appendix), where the critic of PPO is trained with C51. C51 is usually considered to improve representation learning by implicitly providing auxiliary tasks (Bellemare et al., 2017; Munos, 2018; Petroski Such et al., 2019). Figure 14 shows that training the critic with C51 indeed leads to a performance improvement and PPO-C51 ($\gamma_{\mathrm{C}} = 1$) sometimes outperforms PPO-TD ($\gamma_{\mathrm{C}} < 1$) by a large margin. Figure 15 further shows that when $V_{\max}$ is optimized for PPO-C51, the benefit for using $\gamma_{\mathrm{C}} < 1$ in PPO-C51 is less pronounced than that in PPO-TD, indicating the role of $\gamma_{\mathrm{C}} < 1$ and distributional learning may overlap. Figures 6, 7, & 9, suggest that the overlapping is representation learning.

### C.2  OTHER COMPLEMENTARY RESULTS

Figure 16 shows how PPO-TD-Ex ($\gamma_{\mathrm{C}} = 0.995$) reacts to the increase of $N$. Figure 17 shows the unnormalized representation error in the MRP experiment. Figure 18 shows the average episode length for the `Ant` environment in the discounted setting. For `HalfCheetah`, it is always 1000.

## D  LARGER VERSION OF FIGURES

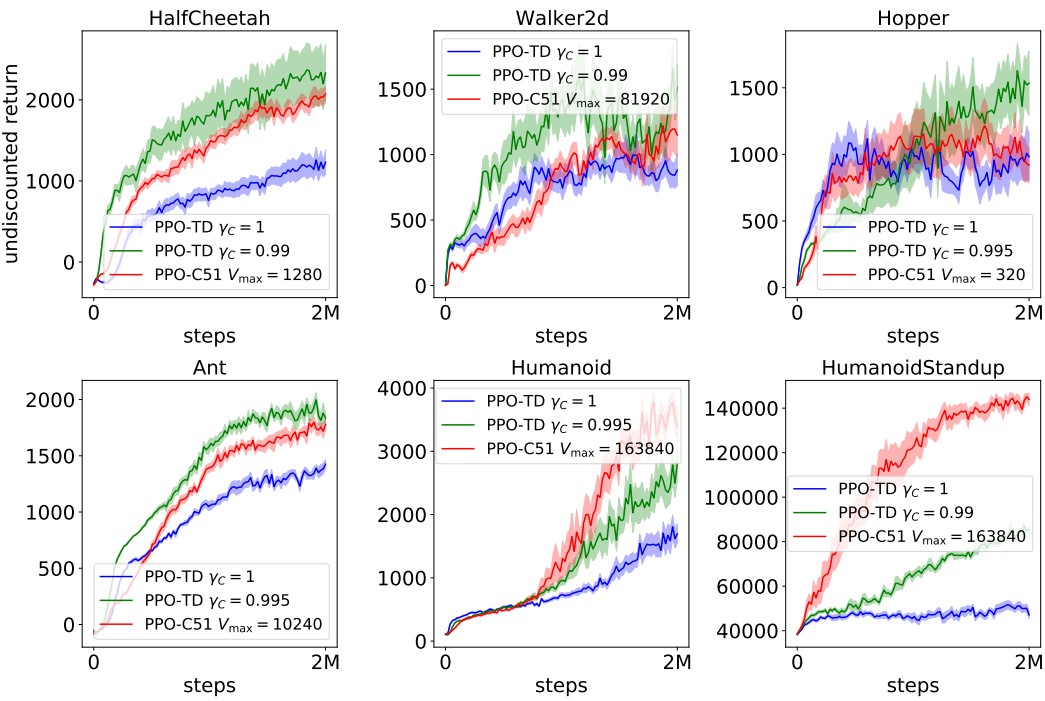

Figure 14: For PPO-C51, we set $\gamma_c = 1$.

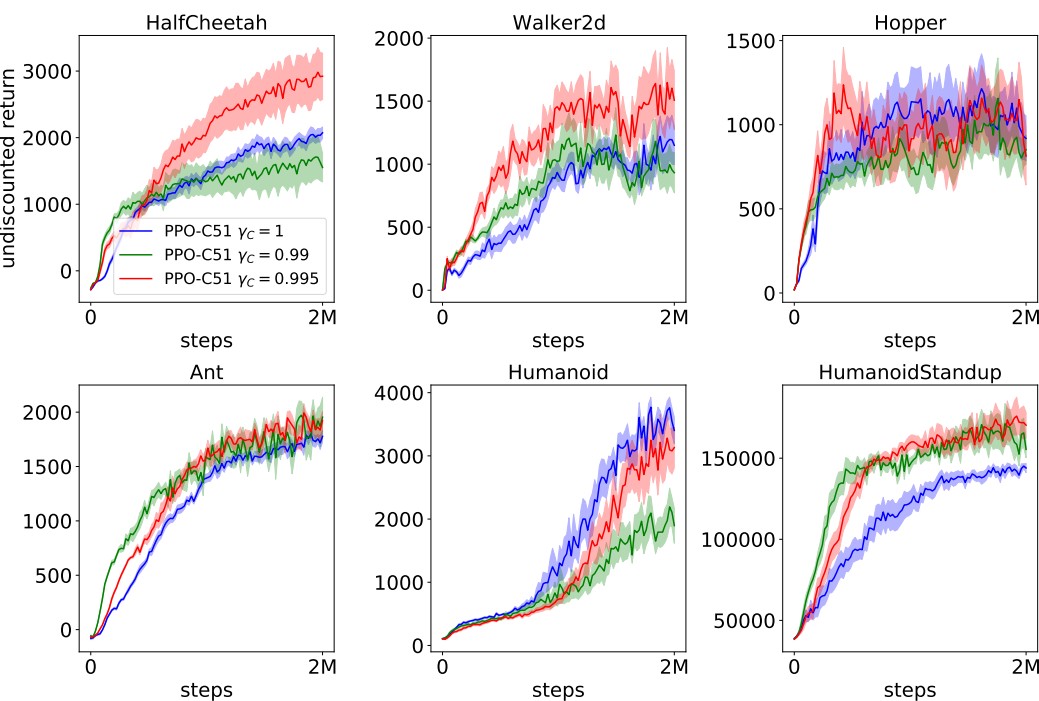

Figure 15: For each game, $V_{\max}$ is the same as the $V_{\max}$ in Figure 14.

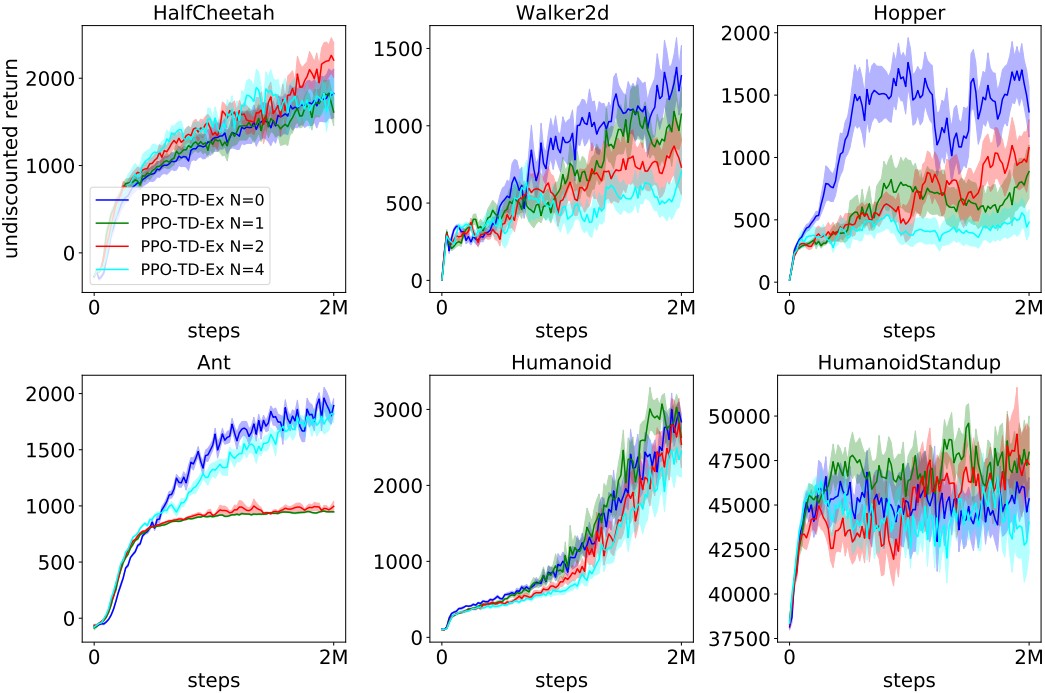

Figure 16: PPO-TD-Ex ($\gamma_\text{c} = 0.995$).

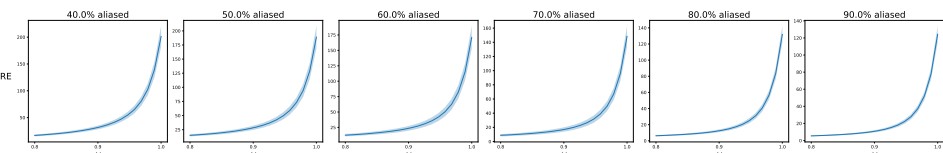

Figure 17: Unnormalized representation error (RE) as a function of the discount factor. Shaded regions indicate one standard derivation. RE is computed analytically as $\text{RE}(X, \gamma) \doteq \min_w ||Xw - v_\gamma||_2$

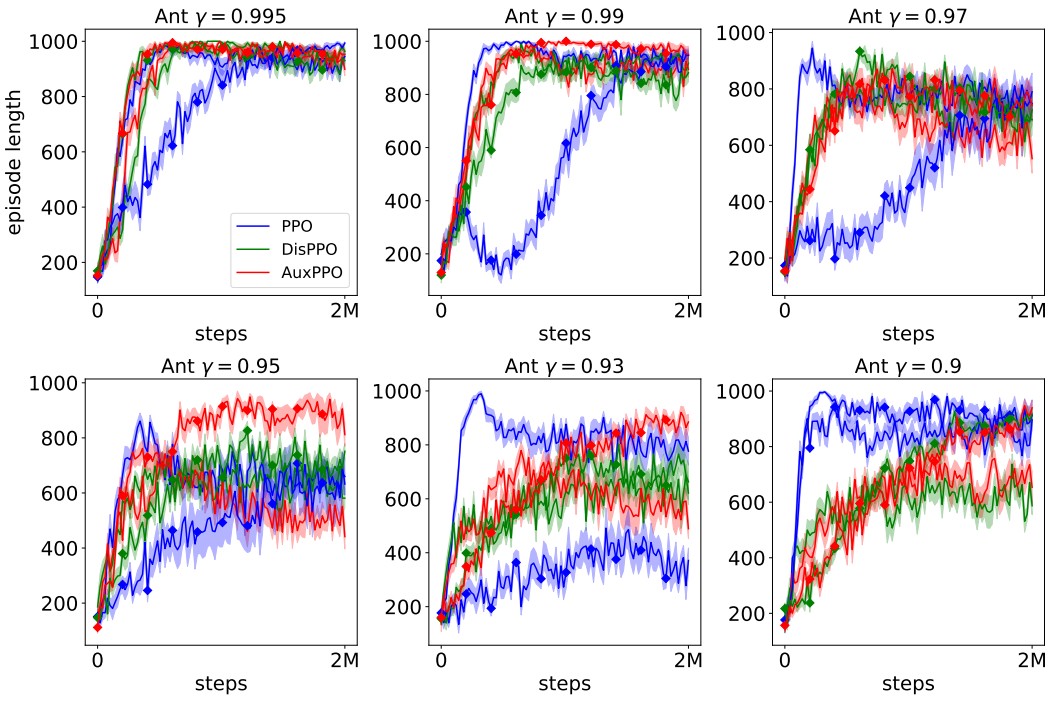

Figure 18: Curves without any marker are obtained in the original `Ant`. Diamond-marked curves are obtained in `Ant` with $r'$.

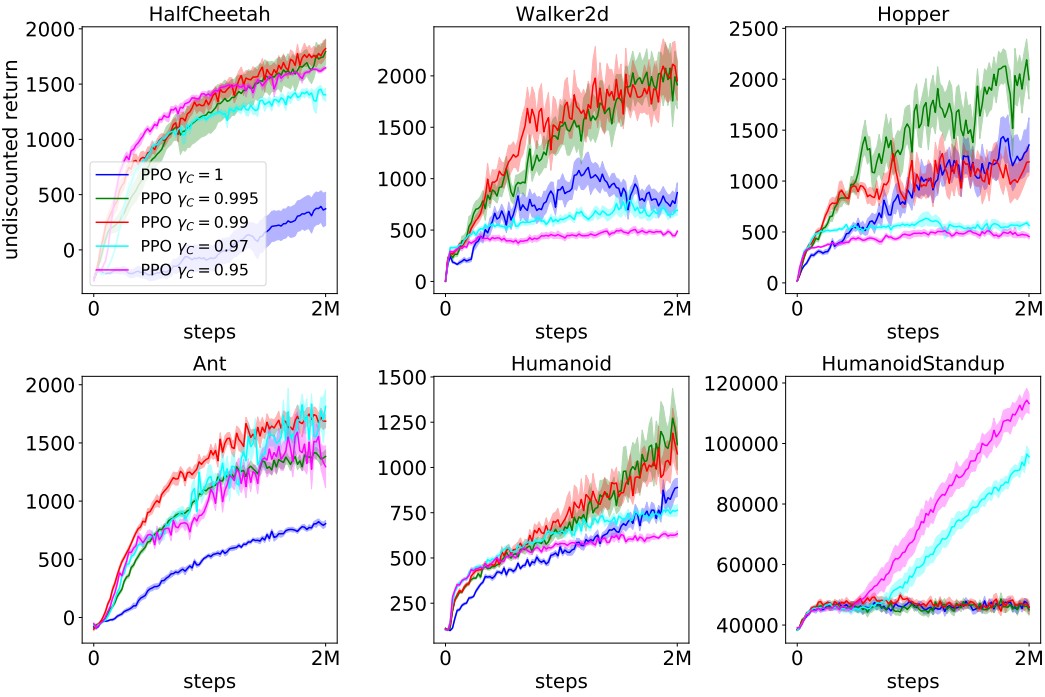

Figure 19: The default PPO implementation with different discount factors. The larger version of Figure 1.

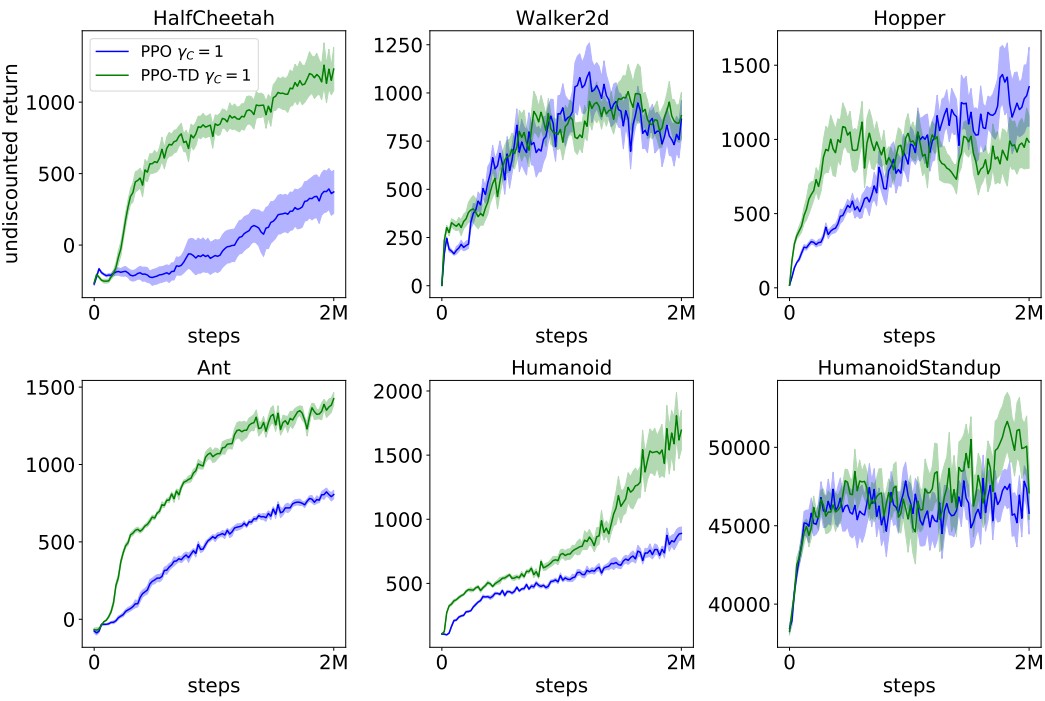

Figure 20: Comparison between PPO and PPO-TD when $\gamma_c = 1$. The larger version of Figure 2.

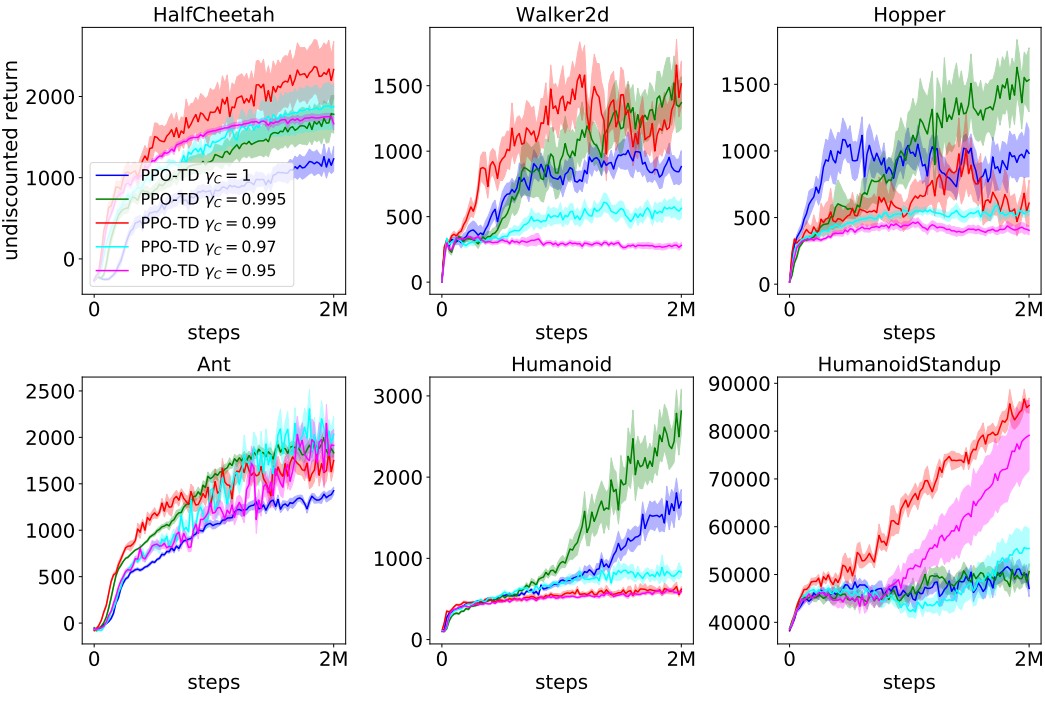

Figure 21: PPO-TD with different discount factors. The larger version of Figure 3.

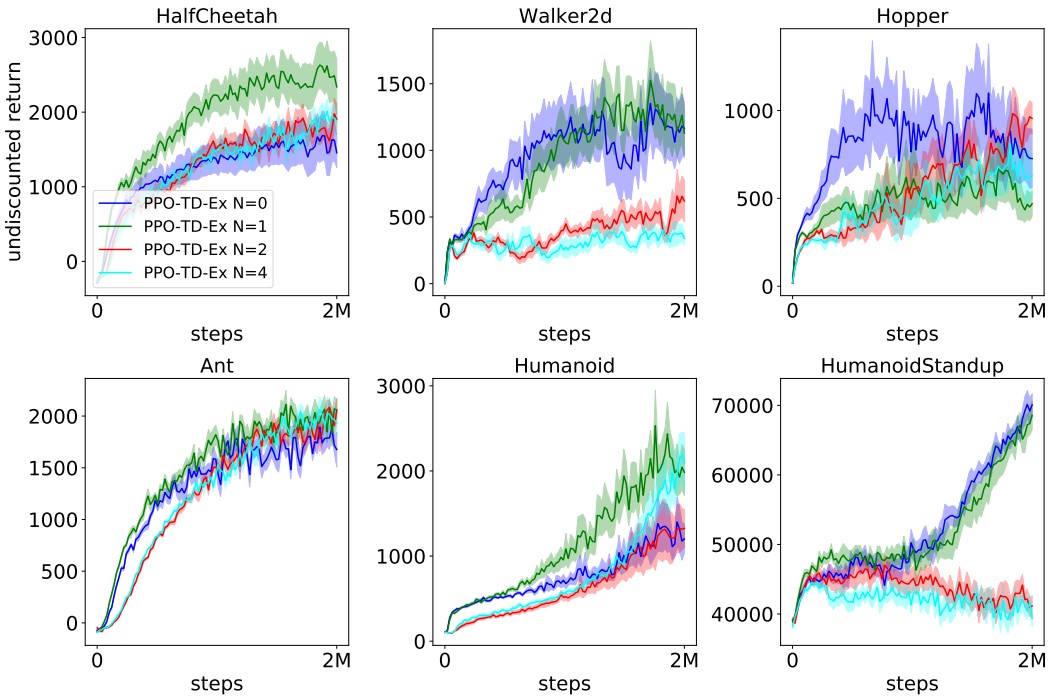

Figure 22: PPO-TD-Ex ($\gamma_c = 0.99$). The larger version of Figure 4.

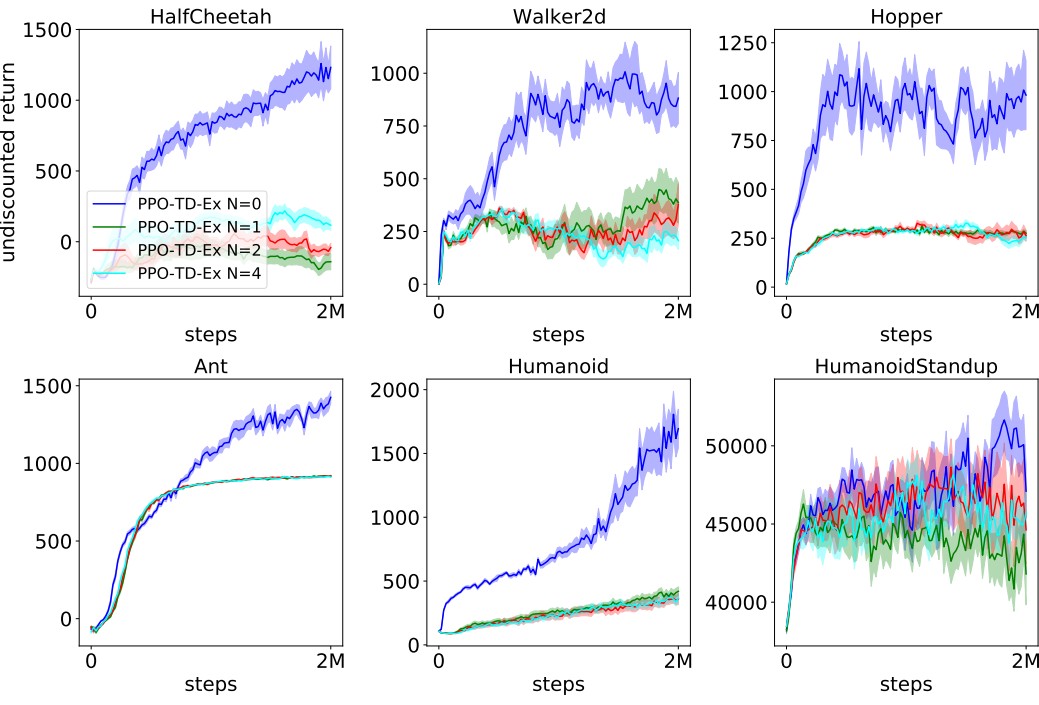

Figure 23: PPO-TD-Ex ($\gamma_c = 1$). The larger version of Figure 5.

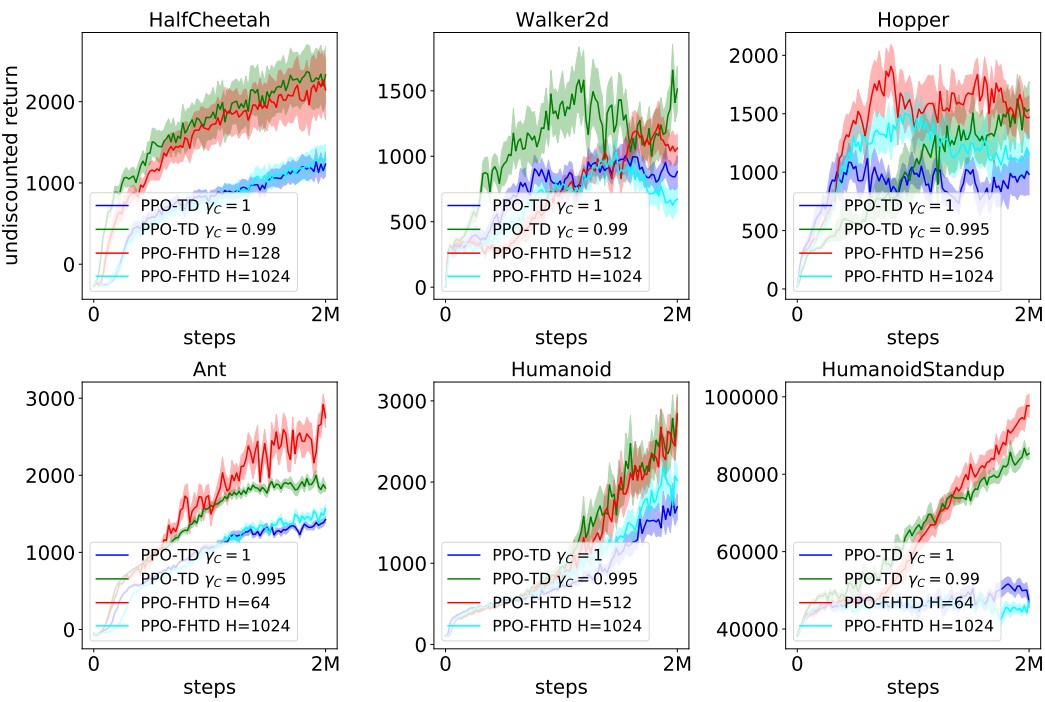

Figure 24: PPO-FHTD with the first parameterization. The best $H$ and $\gamma_c$ are used for each game. The larger version of Figure 6.

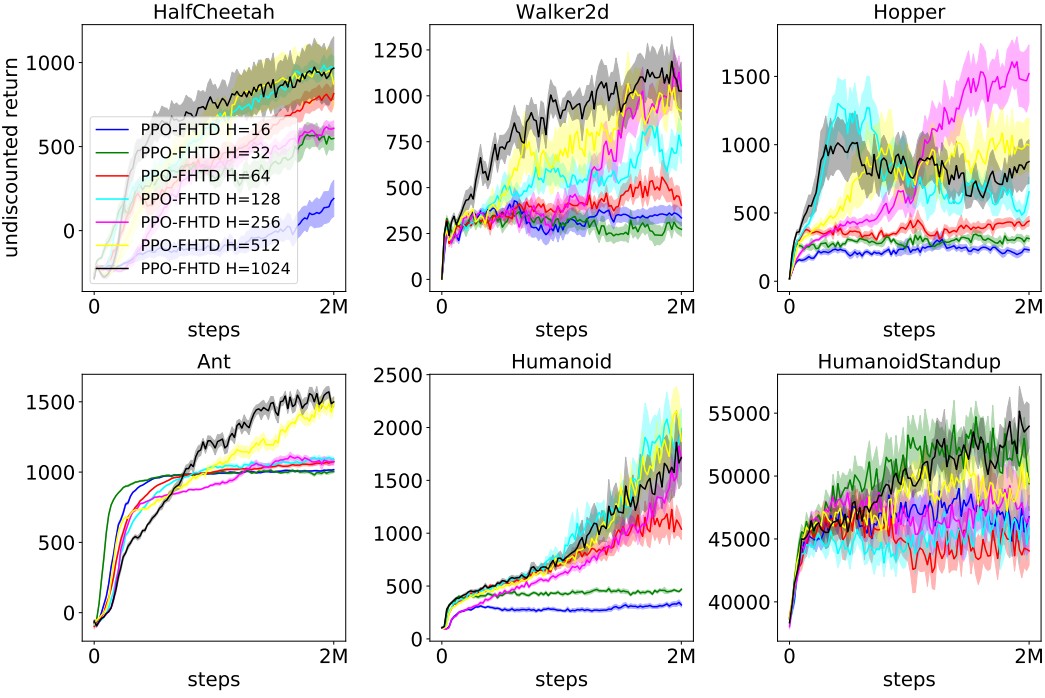

Figure 25: PPO-FHTD with the second parameterization. The larger version of Figure 7.

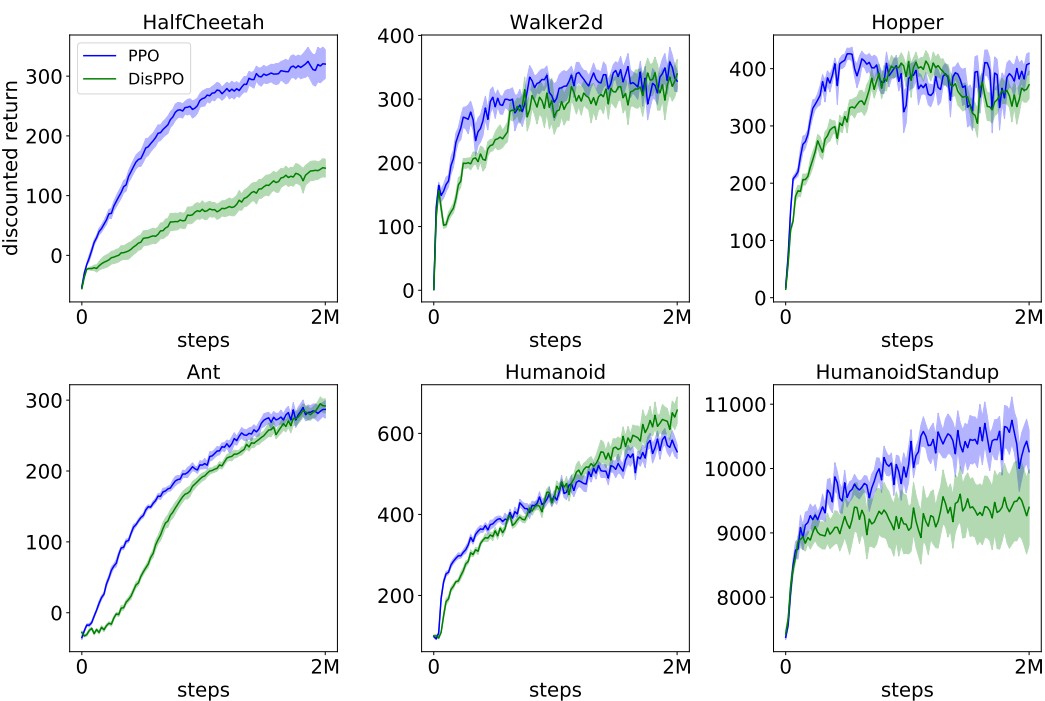

Figure 26: Comparison between PPO and DisPPO with $\gamma = 0.995$. The larger version of Figure 10.

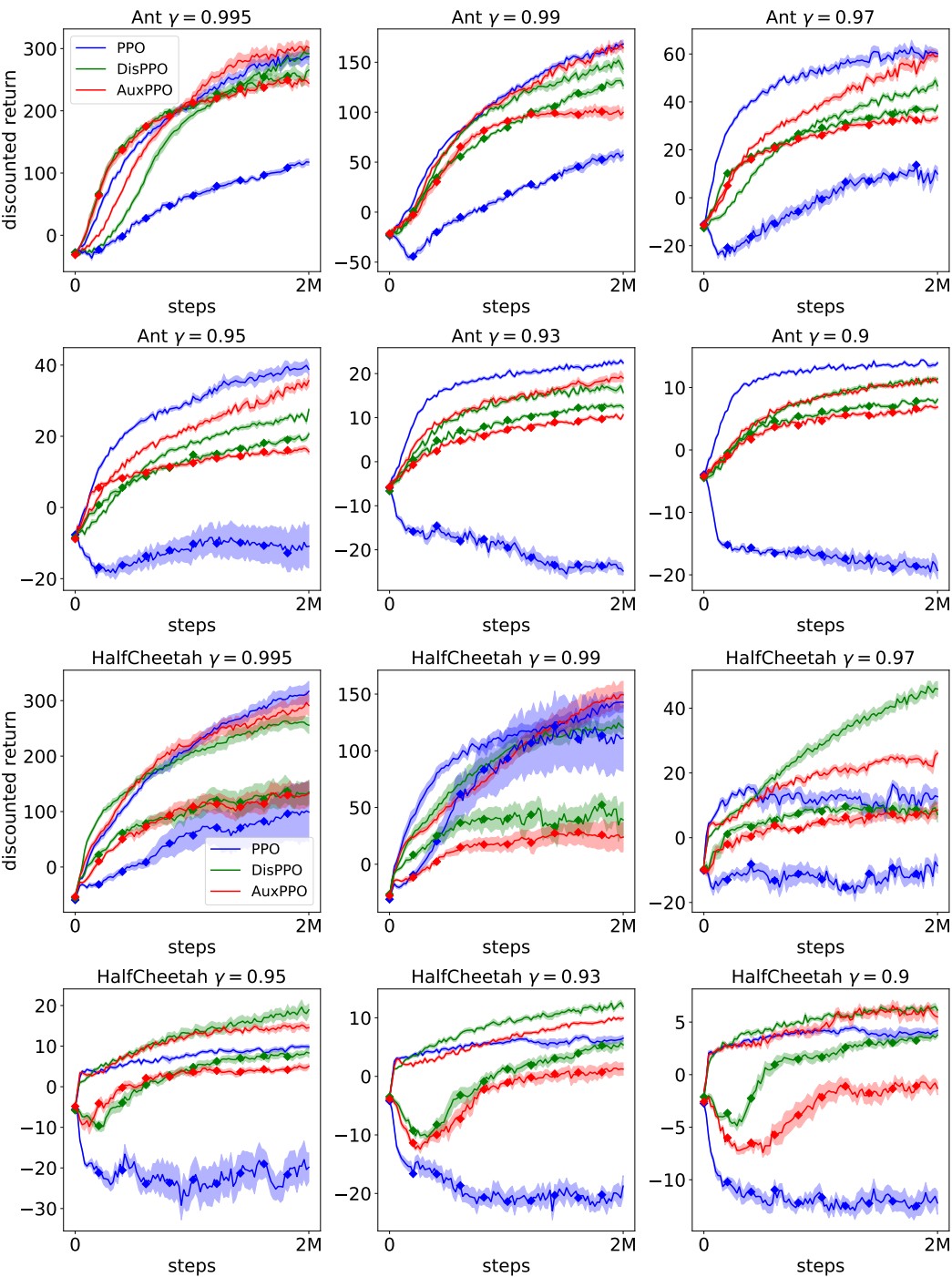

Figure 27: Curves without any marker are obtained in the original `Ant` environment. Diamond-marked curves are obtained in `Ant` with $r'$. The larger version of Figure 11.

