# OpenReview forum: "A Deeper Look at Discounting Mismatch in Actor-Critic Algorithms"
_ICLR.cc/2021/Conference — Reject_

### Official Review · AnonReviewer2 · 2020-10-24
**Good experimental paper with interesting hypothesis, but more analysis expected [Borderline Accept]**

**Rating:** 6
**Confidence:** 4

**Review:**

In this paper, the authors focus on the discounting mismatch in the Actor-Critic algorithm. From comprehensive experiments, the authors claim that this mismatch is either a bias-variance representation tradeoff or an auxiliary task for the actor update. Since the discounting mismatch problem is a well-known gap between the theoretical analysis and the application, their work, especially the experiments, might have some impact on how to understand this gap.

However, since it does not provide any new analysis technique or practical model to improve the performance of the AC algorithm. I would encourage the authors to do more analysis of the choice of $\gamma$, like how to choose $\gamma$ might lead to a good performance (either experimentally or theoretically). And I believe that would have more impact on both the theoretical analysis and practical algorithm design. and Meanwhile, since in the first scenario, the mismatching of $\gamma$ is considered to reduce the variance, it would be interesting if the authors could compare this kind of variance reduction with the stochastic variance reduction on the policy-gradient algorithms [1] [2] [3].

Therefore, though this paper lacks a theoretical analysis or a ground-breaking experimental performance, this paper has an interesting and comprehensive experimental survey and proposes some new hypothesizes on this problem, I will suggest borderline accept for this paper. I might consider modifying my suggestion after discussion with other reviewers and the author's response.

[1] Papini, Matteo, et al. "Stochastic variance-reduced policy gradient." arXiv preprint arXiv:1806.05618 (2018).

[2] Xu, Pan, Felicia Gao, and Quanquan Gu. "Sample efficient policy gradient methods with recursive variance reduction." arXiv preprint arXiv:1909.08610 (2019).

[3] Yuan, Huizhuo, et al. "Stochastic Recursive Momentum for Policy Gradient Methods." arXiv preprint arXiv:2003.04302 (2020).

---

> ### Author Response · Authors · 2020-11-16
> **Response to R2**
>
> Thanks for the constructive feedback. We agree an analysis about the choice of $\gamma$ could potentially increase the impact of this work. However, given the page limit, we decided to leave it for future work. We have included discussion about [1][2][3] in the updated document. However, as we state in the document, the bias-variance trade-off is known in previous work and is not our main contribution. Our contribution is to show that representation learning is at play as well. Based on this, we leave the comparison with other variance reduction techniques for future work.

---

### Official Review · AnonReviewer1 · 2020-10-25
**A Deeper Look at Discounting Mismatch in Actor-Critic Algorithms**

**Rating:** 4
**Confidence:** 3

**Review:**

***Summary***
The paper proposes an empirical study of the discount factor as a regularization parameter in the actor-critic architectures. Specifically, the paper considers the case in which the actor and the critic employ different values of the discount factor. Two scenarios are considered. First, the paper analyzes the case in which the true objective is undiscounted and a discount factor is employed in the critic (like in TRPO and PPO). Second, the case in which the true objective is actually discounted but the discount factor is ignored in the update of the actor. A quite large suite of experimental results is reported.

***Major issues***
- (Organization) The paper presents an extensive experimental evaluation that is split between the main paper and the appendix. However, in the main paper, there are a lot of references and discussions related to experimental results that are provided in the appendix only. This happens both in Section 3 and in Section 4. Sometimes these results (presented in the appendix only) seem to be some fundamental claims of the paper, like for Figures 11, 12, and 13. I think this choice makes affects negatively the readability and clarity of the paper. Indeed, the reader has to continuously jump between the main paper and the appendix. Similarly, the pseudocodes are reported in the appendix only, but I think that this is less relevant compared to the plots. I think that the paper would greatly benefit from a reorganization, making it more self-contained.
- (Bias-Representation Trade-off) One of the main claims of the paper is that using a discount factor < 1 in the critic when the true objective is undiscounted has a regularization effect not only on the variance but also on the learnability of the value function itself. I have to admit that the paper has not convinced me on this point. It is hard to say that the representation of the value function becomes more complex as the discount factor approaches one or, similarly, as the horizon increases. In general, I think that is possible to devise MDPs in which the value function representation becomes simpler as the horizon increases as well as MDPs in which it becomes more complex. I can imagine that for a class of tasks the statement can be true, but the paper does discuss the properties of these tasks. Can the authors elaborate more on this point?
- (Auxiliary task perspective) The paper proposes a perspective of the critic update without a discount factor for a discounted objective as a sum of two terms. However, I have some concerns about the application of the clipping technique independently for the two terms. Why not perform the clipping just once to the original discounted objective?

***Minor issues***
- In Section 2, the MDP model is introduced assuming finite state-action spaces. Is this assumption really necessary? The experiments are carried out on Mujoco tasks that are characterized by continuous state-action spaces.
- The plots are very small, including the ticks and labels on the axis. Moreover, they are not readable when printing the paper in grayscale. I suggest using different linestyles or markers.

***Overall***
I think that the paper addresses a relevant problem that is surely important to bridge the gap between theory and practice. However, I have some concerns about the organization and about the conclusions (especially regarding the bias-representation trade-off) that the paper draws from the presented results. For these reasons, I think that the paper is currently not ready for publication at ICLR.

---

> ### Author Response · Authors · 2020-11-16
> **Response to R1**
>
> Thanks for the constructive feedback. In the following we address the concerns one by one in the same order as they are raised.
> 1. We have moved all important figures to the main text (thanks to the additional one page in rebuttal). It should now be self-contained.
> 2. We do not mean to claim that there is a monotonic relationship between the discount factor/horizon and the difficulty of representation learning. Our goal is to show that representation learning is an important factor that is at play in the misuse of discounting in actor critic algorithms, which is not recognized in previous work to the best of our knowledge. We have explicitly clarified this in the updated document to avoid any confusion.
> 3. The two heads represent two different policies and we therefore have two importance sampling ratios. We think it makes more sense to clip them individually.
> 4. This is mainly for the policy improvement proof of undiscounted TRPO. We use the perturbation-based method for the proof of undiscounted TRPO, which requires a finite MDP. We can also use the coupling technique for a proof with arbitrary MDPs, this will however complicate the presentation and deviate from the main idea of the paper.
> 5. We have included the larger version of all figures in the appendix in the updated document to improve readability.

---

> > ### Comment · AnonReviewer1 · 2020-11-21
> > **Response to Authors**
> >
> > Thank you for your reply. I appreciate the paper's reorganization (especially moving the plots in the main paper) and the provided clarifications. However, considering the other reviewers' issues on experiments' weaknesses, I think that the paper is still not ready for publication. Thus, I keep my score.

---

> > > ### Author Response · Authors · 2020-11-24
> > > **Response to R1**
> > >
> > > Thanks a lot for your updates. It would be much appreciated if you could also point out a particular issue that you are not fully convinced by our response (to other reviewers) so we can improve upon.

---

### Official Review · AnonReviewer3 · 2020-10-27
**A lot of interesting ideas with poor justification**

**Rating:** 4
**Confidence:** 4

**Review:**

The authors examine the commonly used paradigm of not discounting in the policy gradient objective. They propose two hypotheses relating to discounting. (1) discounting the critic improves representation learning. (2) undiscounted policy gradient is similar to discounting + an auxiliary loss. These hypotheses are studied through a series of empirical tests in the MuJoCo domain with PPO.

Strengths:
- I believe this paper is asking the right type of questions about common setups. There are a lot of choices made in deep RL algorithms which don't align with theory and are otherwise unstudied and empirical studies are an important.
- Some of the approaches used to answer these questions are quite unique.
- Overall, there a lot of experiments both in the paper and the appendix, which is detailed. This is a paper which will benefit from the additional page of content as a lot of key figures can be shifted to the main body.

Weaknesses:

Given the empirical nature of this study, it is really important to have robust experimentation to really answer the hypotheses the paper raises. I think the paper falls short at this aspect and I wasn't convinced by the arguments made for either hypothesis. Furthermore, the conclusions that could be drawn from the results are generally not that surprising.
- I'm not sure PPO is the best algorithm to analyze many of these questions. For example, Engstrom et al., 2019 showed a lot of very minor implementation level details had a large impact on the performance. Consequently, it may be difficult to disentangle the actual causative factors in performance. This is problematic as many of the claims in the paper are supported by empirical tests where the performance is not strikingly different. For example, Figure 1 is meant to justify that for $\gamma_c = 0.99$ additional transitions improved performance, but on several environments increasing $N$ to 2 or 4 seems to hurt performance, going against our intuition about variance reduction. Figure 2 shows that for $N \neq 0$ there is a large performance drop, but all values of $N \neq 0$ achieve a very similar performance rather than trending downwards as $N$ increases. To me this suggests a very brittle algorithm.
- For section 3 the bias-variance trade-off is evident from prior work (as referenced by the authors) so the result is of course not novel. I think analyzing it in a deep RL setting is important but because of the problems mentioned prior, I didn't find that these results provide anything solid to add to our understanding.
- The results for Figure 3 aren't convincing (1) because they are overfit, by selecting the best possible H for each it seems likely to always arrive at a high performing agent. (2) This more suggests that these environments don't require the full horizon to achieve a high performance. Consider a simple cartpole problem which is optimal using greedy actions but has a horizon of 1 million time steps. Since were in an approximate setting with deep networks, it isn't surprising that the agent can achieve a high performance without considering the full horizon.
- The results from the toy MRP experiment and distributional RL do suggest some kind of connection to representation learning, but isn't considering a longer horizon simply a more difficult learning problem? Is the representation necessarily an important aspect here? I didn't find that the authors answered this question.
- The conclusion from Section 4 is that $\gamma_A=1$ is an inductive bias that all transitions are equally important seems entirely self-evident from the mathematical definition given it applies equal weight to all transitions. At the same time the main question of hypothesis 2 seems unanswered. Shouldn't AuxPPO $\approx$ PPO, rather than DisPPO if this was true?
- A single environment for Figure 9 is not enough to draw any meaningful conclusions. I did not find the discussion in B.1. convincing that the other environments were not suitable. Simply change $t_0$ for the other environments. From personal experience the horizon of Ant is generally large (near 1000) as the terminal condition is hard to achieve meaning the difference between Ant and the fixed length environments should be small.

Additional Comments:
1. I do wonder if this paper is better off as two separate documents where each hypothesis is provided much more significant attention/experimentation. For example, hypothesis 1 isn't actor-critic specific and is also applicable to Q-learning based methods. These experiments could be simplified by looking at algorithms with significantly fewer components and more settings.
2. For the PPO-TD-Ex experiment I think it's also worth considering extrapolation error (Fujimoto et al., 2019) in TD learning. Since $S^i_{t+1}$ is sampled from a single transition rather than a full trajectory it is not necessarily contained in the batch. As a result, $\hat v$ is not trained on $S^i_{t+1}$ and produces an erroneous TD target. My first impression was that the performance drop for $\gamma_c=1$ was not surprising but the performance gain from $N=1$ for $\gamma_c=0.99$ was, and I think are are unanswered questions here. Another important reference is Bengio et al., 2020 which showed TD(0) generalizes worse than TD($\lambda$) and there is clearly a related result here.
3. Given MuJoCo environments are time-limited to 1000 time steps, 1024 heads for PPO-FHTD seems like a mistake/oversight.
4. Why does PPO-FHTD with H=1024 produce different results for the different parametrizations?
5. Is Figure 6 surprising since the value function needs to consider a large space of solutions as the horizon increases?
6. Given distributional RL provides a large performance gain (which to the best of my knowledge, we are still missing a conclusive reason as to why), I'm not sure PPO-C51 > PPO-TD is a significant result.
7. It would be clearer if DisPPO was described before mentioning Figure 15.
8. Figure 15 seems like an important conclusion and should be contained in the main body of the paper. However, the y-axis of Figure 15 also conflicts with the description in the main body so I'm not sure what the correct interpretation is.
9. I wonder if the result from Figure 9 is reproducible if the flipping was done in a different way. In the MuJoCo environments is the agent is rewarded mainly for velocity and the behavior of the agent in these cases would be enlightening. Does the agent run forward and then attempt to terminate? Can it move backwards?

Conclusion:

I think the authors present a lot of interesting ideas and experimental approaches to answer their underlying questions. However, I felt that the experimentation was not sufficiently robust to justify their conclusions and I cannot recommend acceptance.

References
- Engstrom, Logan, et al. "Implementation Matters in Deep RL: A Case Study on PPO and TRPO." 2019.
- Fujimoto, Scott, et al. "Off-policy deep reinforcement learning without exploration." 2019.
- Bengio, Emmanuel, et al. "Interference and Generalization in Temporal Difference Learning." 2020.

** Edit (Nov 23): I have slightly increased my score due to the improvements made to the paper (mainly reorganization) & some clarifications made by the authors, but I still don't feel like my main concerns were addressed.

---

> ### Author Response · Authors · 2020-11-16
> **Response to R3 (1/2)**
>
> Thanks for the constructive feedback. In the following we address the concerns one by one in the same order as they are raised.
>
> In “Weakness”:
>
> 1. We select PPO mainly because it is a commonly used on-policy baseline in Mujoco domains and Mujoco domains are widely used by deep RL practitioners. To reduce confounders, we tailored PPO for the purpose of our study (e.g., we disabled the GAE part) and conducted thorough hyperparameter tuning. We keep the PPO design choice fixed across all experiments and only draw a conclusion when the performance difference is reasonably significant.
> Sorry we did not make it clear but Figure 4 (previously Figure 1) is NOT meant to justify that additional transitions increase the performance. The goal of Figures 4, 5, and 16 is to study how increasing N interplays with different discount factors. The message we want to convey is that for $\gamma=1$, increasing N hurts more compared with $\gamma = 0.995$ and $\gamma=0.99$. Increasing N will theoretically reduce variance. However, as you mentioned, the critic is not trained on the additional successor states sampled from the oracle. If the critic were to generalize well, the reduced variance should lead to a performance improvement. Otherwise, a performance drop is expected as the value prediction on those additional states can be erroneous. The comparison between Figure 4 and Figure 5 indicates that with $\gamma=0.99$, the critic is likely to generalize better than with $\gamma = 1$, which motivates the study of representation learning. We have clarified this in the updated document.
> Bengio et al., 2020 study the effect of $\lambda$ in generalization; we study the effect of $\gamma$ in representation learning in the context of the misuse of discounting in actor critic algorithms. We have included a discussion about this in the updated document.
> 2. We agree that the bias-variance trade-off from $\gamma$ is evident. Our contribution is to show that there is more than a bias-variance trade-off. Representation learning is at play as well. We have clarified this in the updated document.
> 3. (1) Note that in Figure 6 (previously Figure 3) we also selected the best $\gamma$ for each task. (2) Even if the environments don't require the full horizon to achieve high performance, we still need to figure out what the disadvantages are for using a full horizon. Our experiments show that variance is not the only problem for using a full horizon, representation learning is at play as well. We are not aware of existing literature showing this in the deep RL settings.
> 4. The MRP experiment is purely a supervised learning experiment as the ground truth value is analytically computed for the estimator to regress. There is no credit assignment involved. The conventional wisdom about the difficulty of long horizon problems is mainly about credit assignment, which is not involved in our MRP experiment. The MRP experiment is all about representation instead of credit assignment.
> 5. If the inductive bias that “all transitions are equally important” is correct (in the original environments), then we expect AuxPPO to behave similarly to PPO. If that inductive bias is incorrect (in the environments with flipped rewards), then we expect AuxPPO to behave similarly to DisPPO, as the auxiliary head in AuxPPO does not disturb the control head directly. As shown by Figure 11, both expectations are satisfied.
> 6. In the discounted setting, the discount factor is part of the problem definition. Though we had a single environment, we used 6 discount factors, leading to 6 different tasks. To further support our hypothesis, we include new experiments in HalfCheetah as well in the updated document (Figure 11), with a thorough learning rate search. The overall conclusion is unchanged. We also updated the discussion about Figure 11 in the updated document. Moreover, the horizon of Ant is not always large. We have included Figure 18 in the appendix in the updated document showing the average episode length of Ant in different settings.

---

> > ### Author Response · Authors · 2020-11-16
> > **Response to R3 (2/2)**
> >
> > In “Additional Comments”:
> > 1. Thanks for this constructive suggestion.
> > 2. Please see “1” in the reply to weakness
> > 3. As shown by Figures 6 and 7, the two parameterizations of PPO-FHTD with H=1024 indeed have very similar performance (there are differences resulting from random seeds). At the first glance they look different because the range of the y-axis is different in the two figures.
> > 4. The representation error in Figure 9 (old Figure 6) is normalized by the scale of the true value function. Considering a large space of solutions is indeed one potential reason why a better representation is needed. The goal of this work is to show that representation learning is also at play (beyond bias-variance trade-off) in the misuse of the discounting in actor-critic algorithms.
> > 5. We have moved the distributional RL experiments to the appendix. We agree there is no unanimously agreed explanation for the improved performance of distributional RL.
> > 6. We moved Figure 10 (previously Figure 15) to the main text in the updated document, which should make the use of DisPPO less abrupt.
> > 7. Thanks for pointing this out. This is a typo. The correct y-axis should be "discounted return". We have fixed this in the updated document.
> > 8. When we flip the reward, the Mujoco agent can run backwards. This feature has been used in previous literature (see, e.g., Zhang et al. (2019); Hafner et al. (2019)). The recordings of the agent behavior are also available there.
> >
> > Zhang, S., & Whiteson, S. (2019). DAC: The double actor-critic architecture for learning options. In Advances in Neural Information Processing Systems (pp. 2012-2022).
> >
> > Hafner, D., Lillicrap, T., Fischer, I., Villegas, R., Ha, D., Lee, H., & Davidson, J. (2019, May). Learning latent dynamics for planning from pixels. In International Conference on Machine Learning (pp. 2555-2565). PMLR.

---

> > > ### Comment · AnonReviewer3 · 2020-11-24
> > > **Response to authors' rebuttal**
> > >
> > > Thank you for responding and addressing some of my concerns. I appreciate the effort put into updating the paper. I have increased my score slightly, but I still recommend rejection, as I don’t feel the experiments are sufficiently robust to justify the main hypotheses.
> > >
> > > Additional comments after re-reading:
> > > - Figure 9 (MRP) uses a linear function with an approximation to state aliasing. Since the objective is to show this with neural networks, why not just use a neural network directly? At the same time, it's unclear why this experiment captures an aspect of "representation learning", the NRE metric is just a normalized Euclidean distance, is this really saying anything about representation learning?
> > > - If the authors are truly convinced by their experiments then I suspect a major issue with the paper is clarity. A lot of the experiments do not directly demonstrate their proposed idea and following the authors justification is not always straightforward. Taking Figure 9 as an example, I feel like I'm missing the explanation that closes the gap between the experiment and the conclusion of bias-representation.
> > > - Figure 11: the diamond markers are indiscernible without zooming in greatly.

---

> > > > ### Author Response · Authors · 2020-11-24
> > > > **Response to R3**
> > > >
> > > > Thanks a lot for your updates.
> > > > We want to clarify that the MRP is mainly about "representation" instead of "representation learning". There is no learning in the MRP experiment. The goal of the MRP experiment is to empirically show that with certain discount factor (in the MRP case, a smaller discount factor), good representations can be much more dense. We use linear features because we interpret representation learning as learning the backbone of the network. The last layer is then interpreted as a linear function approximator.

---

### Official Review · AnonReviewer4 · 2020-10-28
**This work investigates an open problem in RL relating to the missing discount factor in actor-critic algorithms. While this problem has been studied from a theoretical perspective in the past, this work focuses on an empirical analysis instead.**

**Rating:** 6
**Confidence:** 4

**Review:**

Overall I like this direction since this is an important, open problem in RL that does not seem to be widely known (I was unaware of it until I looked into the related work) and could lead to improved algorithms. I encourage the authors to continue to pursue this line of research. However, I have a few clarifications and questions regarding the experiments which make it unclear how meaningful the results are. For now, I vote to reject this work but am willing to change my opinion based on the rebuttal.

Strengths:
- The paper investigates and draws further attention to an important open problem that does not seem to widely known. Based on my reading of Nota and Thomas, it appears most major papers in the field today do not acknowledge the discrepancy of the missing discount factor.
- The paper includes many experiments especially in the Appendix each with a robust 10 seeds. I do have some issues with the experimental setup that I will detail later but I appreciate the variation in experiments.
- I also think the representation learning experiments in Scenario 1 using FHTD are an interesting approach to study the effect of learnt representations.
- The experimental setup and methods used are clearly described and it appears the code will be made available in the final version thereby potentially making the experiments highly reproducible.

Issues/Points of clarification:
- Most of the study is done in the setting where \gamma=1 (Scenario 1 in the paper). This corresponds to the undiscounted objective where the current time index must be included in the state for correct estimation of the value function. However the setting that is most widely used in existing literature involves a discount factor<1. For instance, all of the methods cited in the Methodology section: Henderson et al., 2017; Ilyas et al., 2018; Engstrom et al., 2019; Andrychowicz et al., 2020, Fujimoto et al., 2018, Haarnoja et al., 2018 use a discount<1 (Andrychowicz et al. do not include a discount of 1 in their sweep over discount factors either).  This is dubbed Scenario 2 in the main text and includes only one experiment on the Ant task. It is fine to try to draw insights and focus on Scenario 1 as long as it is well motivated. However I do think it is misleading to claim ‘we believe our empirical results are relevant to most practitioners’ when most of the study does not involve a setting that is actually used by said practitioners.

- My second concern is with the method used to choose hyperparameters for the experiments. In particular, the learning rate is chosen based on the ‘Ant’ experiment and then the best performing parameters are fixed and transferred to the others. While I appreciate the motivation behind this approach, I’m not certain how well these transfer to some tasks. In particular, the HumanoidStandup task seems to involve returns which are an order of magnitude greater than the other tasks. I think at least for this one task a small sweep is essential to be confident of the claims.

- There are a few points in the paper where correlation seems to be misinterpreted as causation. For instance Figures 11-13 in the paper indicate that: a) a discounted critic (\gamma_c<1) performs better on all tasks; b) biased updates using TD instead of empirical returns performs better on some tasks. These two statements alone are insufficient to claim that the advantage of a discounted critic (\gamma_c=1) is therefore partly due to bias. Looking at Figures 11 and 13, I think a figure similar to Figure 12 comparing TD and empirical returns can be generated for any discount factor (e.g. \gamma=0.995). Perhaps I am missing something here and if so clarification from the authors would be much appreciated!

- These discrepancies combine in Figure 1 where for \gamma_c=0.99, different values of extra transition samples (N) are plotted. Ostensibly, increasing N should reduce the variance even further. However quite a few of the curves choosing N=2 or 4 performs significantly worse. Could the authors clarify why they think this happens? Interestingly, the only task where the effect of N seems to not matter is the Ant task for which a hyperparameter sweep was completed. Additionally the task where increasing N impacts performance the most is the HumanoidStandup task where the returns are quite significantly different. To me, this result stresses that there might be more at play here and a more detailed study is required to tease apart the various confounding factors.

In summary, while I think the approach is quite interesting, there are concerns in some of the claims made in the text. I appreciate the effort that went into the current set of results and the experimental setup. With that in mind, I would be willing to accept this submission if my concerns above are clarified and if the conclusions drawn from the results are tempered given the evidence.

Finally there are minor points of clarification that did not affect my overall review but I nonetheless list below:

- In the discounted infinite horizon setup of Scenario 2, the timestep no longer needs to be added to the state. However the text indicates that this is still done even in this case. I think this does affect bootstrapping and thus learning the value target. Specifically it may be easier to learn a consistent value function that in this setting when the time index is not included in the state. Could the authors clarify this point?

- As a minor point for readability, it would be good if the algorithm boxes for PPO-TD and PPO-TD-Ex etc included colours to highlight the changes to PPO (Algorithm 1) since these overlap quite a bit. This is purely from a presentation perspective of course.

References:
Peter Henderson, Riashat Islam, Philip Bachman, Joelle Pineau, Doina Precup, and David Meger. Deep reinforcement learning that matters. arXiv preprint arXiv:1709.06560, 2017.

Andrew Ilyas, Logan Engstrom, Shibani Santurkar, Dimitris Tsipras, Firdaus Janoos, Larry Rudolph, and Aleksander Madry. A closer look at deep policy gradients. arXiv preprint arXiv:1811.02553, 2018.

Logan Engstrom, Andrew Ilyas, Shibani Santurkar, Dimitris Tsipras, Firdaus Janoos, Larry Rudolph, and Aleksander Madry. Implementation matters in deep rl: A case study on ppo and trpo. In International Conference on Learning Representations, 2019.

Marcin Andrychowicz, Anton Raichuk, Piotr Stanczyk, Manu Orsini, Sertan Girgin, Raphael ´ Marinier, Leonard Hussenot, Matthieu Geist, Olivier Pietquin, Marcin Michalski, et al. What ´ matters in on-policy reinforcement learning? a large-scale empirical study. arXiv preprint arXiv:2006.05990, 2020.

Scott Fujimoto, Herke van Hoof, and David Meger. Addressing function approximation error in actor-critic methods. arXiv preprint arXiv:1802.09477, 2018.

---

> ### Author Response · Authors · 2020-11-16
> **Response to R4**
>
> Thanks for the constructive feedback. In the following, we address the concerns one by one in the same order as they are raised.
> 1. The cited works (as well as most deep RL papers) use undiscounted return as the performance metric when they report their results in plots and compare different algorithms. If the evaluation uses the undiscounted return, their true objective is indeed undiscounted. Consequently, setting $\gamma < 1$ in their algorithms is a practical heuristic. Our scenario 1 studies why this heuristic works.
> 2. To support our method for selecting learning rates, we provide a new experiment in Figure 12 in the appendix, which shows that the best learning rates of Ant achieves reasonably good performance in all the other games except for Humanoid, indicating our method is reasonably robust. In the paper, we do not draw a conclusion from a single task. So an outlier is unlikely to affect the overall conclusion.
> 3. We agree it shows correlation and not necessarily causation and we have toned our claim down in the updated document. The bias-variance trade-off comes mainly from the math directly. Consider $G_t^\gamma = \sum_{t=0}^\infty \gamma^t R_{t+1}$. For an unbiased estimation, we should use $\gamma=1$. When decreasing \gamma, we introduce bias but reduce the variance of $G_t^\gamma$
> 4. Increasing N will indeed reduce variance. However, the critic is not trained on the additional successor states sampled from the oracle. If the critic were to generalize well, the reduced variance should lead to a performance improvement. Otherwise, a performance drop is expected as the value prediction on those additional states can be erroneous. The comparison between Figure 4 and Figure 5 (previously Figures 1 and 2) indicates that with $\gamma=0.99$, the critic is likely to generalize better than that with $\gamma = 1$, which motivates the study of representation learning. We have clarified this in the updated document.
> Moreover, Ant is not the only environment where N does not matter. In Figure 16, we see that N has no impact on HalfCheetah, Humanoid, and Humanoid Standup and actually does matter in Ant. It may be better to read Figures 4, 5, and 16 together to study how increasing N interplays with different discount factors.
> 5. In Mujoco, theoretically the time index should always be included in the state. Mujoco has a hard time limit, so if the state does not contain the time information, the problem becomes partially observable. See Pardo et al. (2018) for a detailed discussion.
> 6. We highlighted the difference of PPO-TD and PPO-TD-Ex in the updated document.
> 7. We toned down the conclusion from the distributional RL experiments in the updated document.

---

> > ### Comment · AnonReviewer4 · 2020-11-24
> > **Response to authors**
> >
> > I thank the authors for their clarifying remarks. The clarifications went a long way in helping me understand the setup better. I was originally under the misapprehension that the paper was considering the time-unlimited task setting as described in Pardo et al. After re-reading the setup, I see that this was because the returns are defined to be infinite-horizon but effectively the setup uses a finite-horizon with an absorbing state. This helps greatly clarify Scenario 1 a few points to me. With this in mind I have edited and increased my original score for this submission. I am also happy to see that the claims have been toned down by some margin in the paper which helped in my decision. Having said that, I still think much can be done in improving this work to increase its impact.
> >
> > While I appreciate the extra effort that went into running the baseline against learning rates, I still do think a small sweep over learning rates is good practice in empirical studies of this kind. I don't mean that each experiment should be run with the significant number of parameters now included in Figure 12, but sweeping over a few parameters for each task can be quite beneficial. However I do see the authors point in that the specific learning rate chosen for their one task seems reasonably robust and in particular works for the HumanoidStandup task that I was concerned about.
> >
> > Finally I think the paper could greatly improve with clarity in presentation. For instance the scale of the Y-axis in Figures 6 and 7 being different made it hard to parse. This really made it much harder to understand the claim - albeit I do agree there may be something here after having spent considerable time staring at this figure. I appreciate that presentation is made difficult with the page limits but I think perhaps the text would read better if some parts (MRP experiment; maybe the whole of Scenario 2?) are moved to the Appendix to allow for more hand-holding for the reader in the first part. Reading some of the other reviews, my impression is readability is possibly a major issue in its current form.

---

### Author Response · Authors · 2020-11-16
**Summary of the revision**

We thank all the reviewers for the constructive feedback. We have posted an updated version of the document to clarify concerns raised in the reviews. In the following, we list the major updates we made in the document.
1. We moved all the important figures in the appendix (Figures 11, 12, 13, 15 in the old version) to the main text so the paper is now self-contained (Thanks to the additional page for rebuttal).
2. We added new experiments (Figure 12 in the appendix) to further justify our methodology for selecting hyperparameters.
3. We added new experimental results (Figure 11) for the discounted setting to further support our hypothesis.

---

### Decision · Program_Chairs · 2021-01-07
**Final Decision**

**Decision:**

Reject

**Comment:**

This paper studies the effect of the discount mismatch in actor-critics: the discount used for evaluation (often 1), the discount used for the critic and the discount used for the actor. There’s notably a representation learning argument supported by a series of experiments.
The initial reviews pointed out that this paper addresses very relevant research questions, sometimes in a quite original way, with a large set of experiments. However, they also raised concerns about the organization/clarity of the paper, and possible weaknesses about the experimental studies.
The authors provided a rebuttal and a revision, that clarified some points and triggered additional discussions. However, if the revision improved the initial submission, the shared assessment is that the clarity and experiments themselves are still somewhat lacking. As such, the AC cannot recommend accepting this paper.
Yet, this work does have interesting ideas, and the problem considered is of interest for the community and under studied. The authors are strongly encouraged to submit a revised version to a future venue.